# A benzodiazepine activator locks K$_v$7.1 channels open by electro-mechanical uncoupling

Julian A. Schreiber [1,2,8], Melina Möller[1,8], Mark Zaydman[3], Lu Zhao[3], Zachary Beller[3], Sebastian Becker[1], Nadine Ritter[1,4], Panpan Hou[3], Jingyi Shi[3], Jon Silva[3], Eva Wrobel[1], Nathalie Strutz-Seebohm[1], Niels Decher [5], Nicole Schmitt [6], Sven G. Meuth[7], Martina Düfer[2,4], Bernhard Wünsch [2,4], Jianmin Cui [3] & Guiscard Seebohm [1,4✉]

Loss-of-function mutations in K$_v$7.1 often lead to long QT syndrome (LQTS), a cardiac repolarization disorder associated with arrhythmia and subsequent sudden cardiac death. The discovery of agonistic $I_{Ks}$ modulators may offer a new potential strategy in pharmacological treatment of this disorder. The benzodiazepine derivative (R)-L3 potently activates K$_v$7.1 channels and shortens action potential duration, thus may represent a starting point for drug development. However, the molecular mechanisms underlying modulation by (R)-L3 are still unknown. By combining alanine scanning mutagenesis, non-canonical amino acid incorporation, voltage-clamp electrophysiology and fluorometry, and *in silico* protein modelling, we show that (R)-L3 not only stimulates currents by allosteric modulation of the pore domain but also alters the kinetics independently from the pore domain effects. We identify novel (R)-L3-interacting key residues in the lower S4-segment of K$_v$7.1 and observed an uncoupling of the outer S4 segment with the inner S5, S6 and selectivity filter segments.

[1] Institute for Genetics of Heart Diseases (IfGH), Department of Cardiovascular Medicine, University Hospital Münster, D-48149 Münster, Germany. [2] Institute of Pharmaceutical and Medicinal Chemistry, University of Münster, Corrensstr. 48, Münster D-48149, Germany. [3] Department of Biomedical Engineering, Center for the Investigation of Membrane Excitability Disorders, Cardiac Bioelectricity and Arrhythmia Center, Washington University, St. Louis, MO 63130, USA. [4] Chembion, University of Münster, D-48149 Münster, Germany. [5] Institute of Physiology and Pathophysiology, Vegetative Physiology, Philipps-University of Marburg, Deutschhausstr. 1-2, 35037 Marburg, Germany. [6] Department of Biomedical Sciences, University of Copenhagen, Copenhagen, Denmark. [7] Department of Neurology, University Hospital Düsseldorf, Düsseldorf, Germany. [8] These authors contributed equally: Julian A. Schreiber, Melina Möller. ✉email: guiscard.seebohm@ukmuenster.de

K$_v$7.1 channels are the founding members of voltage-gated K$_v$7 delayed rectifier potassium channels (K$_v$), that play important roles in various tissues including epithelia, brain, heart, and inner ear organs[1,2]. In native tissues, K$_v$7.1 α-subunits (encoded by *KCNQ1*) can interact with various ancillary subunits or modulatory proteins, which constitute the biophysical properties of the channel producing functionally distinct K$^+$-currents[3–7]. In cardiac myocytes, co-assembling of the pore-forming α-subunit and its regulatory β-subunit KCNE1 (mink, IsK) generates the slow component of the cardiac delayed rectifier potassium current $I_{Ks}$ that is critical for the repolarization of the cardiac action potential[8,9]. Loss-of-function mutations in K$_v$7.1 can lead to prolonged cardiac repolarization and cause long QT syndrome (LQTS), a genetically heterogeneous cardiac arrhythmia that is characterized by a prolonged ventricular repolarization phase[10–12]. Recently, diabetes was linked to *KCNQ1* as well[13–17]. Furthermore, K$_v$7.1 channels are important in many other organs and contribute to different physiological functions[1,2,4]. Thus, K$_v$7.1 channel modulators hold the potential for the development of new pharmacological strategies in the treatment of many diseases like cardiac arrhythmias, diabetes, diarrhea, impaired function of the thyroid gland and others.

Homomeric K$_v$7.1 channels exhibit activation upon membrane depolarization and undergo delayed partial inactivation. This delayed slow inactivation cannot be observed directly upon channel activation but becomes observable when the membrane is stepped to hyperpolarized potentials after extended channel (in) activation and channels switch from inactivated to activated states before closing. A characteristic hook in the tail can thus be seen in the current traces. These characteristics of Kv7.1 gating have been proposed to be the result of at least two open states that need to be occupied during the gating process, from which the channel can enter flicker states with several sub-conductive states[18–20].

The prototypic high potency activator of K$_v$7.1 is the benzo-diazepine derivative (*R*)-L3 (L-364,373), which leads to increased $I_{Ks}$ current amplitude, shortens action potential duration in guinea pig cardiac myocytes and suppresses early after-depolarizations in rabbit ventricular myocytes[21,22]. The K$_v$7.1 channel modulation by (*R*)-L3 is associated with pronounced alterations of gating parameters[21,23]. (*R*)-L3 appears to arrest the channel in closed and open states and therefore prevent the channel from entering the inactivated state, which is appreciable as an absent hook in the tail currents[23]. Thus, the presence of (*R*)-L3 seems to reduce or eliminate a particular gating transition towards sub-conductive states. The β-subunit KCNE1 increases channel conductance, slows channel activation and deactivation and markedly reduces macroscopic inactivation as well[8,9,19,24–27]. Thus, (*R*)-L3 and KCNE1 cause somewhat similar effects on K$_v$7.1 channel.

Given the therapeutic potential of (*R*)-L3-like compounds, a strong interest in defining the compound binding site and understanding its molecular mechanism of action exists. Previous data revealed that specific residues in the K$_v$7.1 pore domain interact with (*R*)-L3 allowing the compound to slow K$_v$7.1 channel kinetics and increase the ion channel current[23]. However, the precise molecular mechanism of channel modulation remained elusive. Some of the previously identified amino acids, that have a high impact on (*R*)-L3 activity, are located at the interface between voltage-sensor and pore domain. Thus, this study suggests that the voltage-sensor domain (VSD) might be an important structural target to develop specific Kv7 channel agonists[28].

We hypothesized that since (*R*)-L3 changes the activation and deactivation kinetics and the voltage-dependency of K$_v$7.1 activation, (*R*)-L3 may also physically interact with the VSD. To address this question, we systematically probed the lower S4 segment of K$_v$7.1 via site-directed mutagenesis and subsequently analyzed mutated channel parameters in the absence and presence of (*R*)-L3. To gain insights into the molecular mechanism of action, we performed experimental data guided 3D-modeling of K$_v$7.1 and (*R*)-L3 to identify the complete binding site. Subsequent molecular dynamics simulations were utilized to assess the stability of drug binding and to explore the binding mode of the drug.

## Results

**(*R*)-L3 activates hK$_v$7.1 channels expressed in *Xenopus* oocytes.** In this study, we extend our analysis of the putative (*R*)-L3 (Fig. 1a) binding site towards the VSD by two-electrode voltage-clamp experiments using hK$_v$7.1 expressing *Xenopus laevis* oocytes (Fig. 1b, c). In agreement with our previous electro-physiological studies, we find a robust K$_v$7.1 agonism by (*R*)-L3[23], which is characterized by an EC$_{50}$ value of 4.2 ± 3.5 μM ($n = 20$; SE) and a maximum activation by 212 ± 55% (SE, Fig. 1d). In accordance with our previous data, (*R*)-L3 shifts the voltage-dependent activation from −20 ± 1 mV (SE) to −28 ± 1 mV for $V_{1/2}$ (SE, Fig. 1e). More pronounced alterations can be observed for the activation and deactivation kinetics (Fig. 1f–i). 1 μM (*R*)-L3 increases the time constants for fast activation ($\tau_{fast\ act.}$) as well as for the slow deactivation ($\tau_{slow\ deact.}$) components leading to slowed activation and deactivation of the channel.

**(*R*)-L3 activation properties depend on ion composition.** The observed effects of (*R*)-L3 result from direct interactions with the K$_v$7.1 channel. To specify if the observed effects are a result from direct interactions with the pore domain (PD), we tested whether changing the permeating ion can influence the modulating properties of (*R*)-L3. Therefore, we conducted recordings using 1 μM (*R*)-L3 with 100 mM K$^+$ and 100 mM Rb$^+$ containing buffers (Fig. 2a, b). Interestingly, the activation of corrected (inward) tail current amplitude was slightly but not significantly reduced in high Rb$^+$ (42 ± 5%, SEM, $n = 18$) compared to K$^+$ (54 ± 8%, SEM, $n = 17$, Fig. 2c). Although direct comparison of effects is difficult since external K$^+$ has effects on the onset of channel inactivation[29], activation of corrected tail current amplitude in ND96 (96 Na$^+$/4 mM K$^+$) is significantly larger (84 ± 7%, SEM, $n = 30$). The previous observed shift of $V_{1/2}$ in the presence of (*R*)-L3 in ND96 is weakened in high K$^+$/Rb$^+$ (Fig. 2d, e). In 100 mM K$^+$ buffer $V_{1/2}$ shifts from −30 ± 1 mV (SE) to −31 ± 1 mV (SE) in the presence of (*R*)-L3, while $V_{1/2}$ in 100 mM Rb$^+$ is reduced from −42 ± 1 mV (SE) to −47 ± 2 mV (SE). On the other hand, the modulation of fast and slow deactivation kinetics in the presence of (*R*)-L3 in high K$^+$/Rb$^+$ buffers are similar to the observed modulation in ND96 leading to unaffected fast and increased slow deactivation time constants (Fig. 2f, g, SI Fig. 1). Summarizing these results, the effects of (*R*)-L3 are modulated by different external ion species implicating a direct effect on the pore domain. Moreover, the results are in accordance with our previous study identified several residues from the pore-forming S5 and S6 helix, that significantly decrease the efficiency of (*R*)-L3 (Fig. 2h)[23]. Taken together, the (*R*)-L3 effects partially depend on a direct influence on the pore domain.

**VSD movement is altered in the presence of (*R*)-L3.** Several previous identified amino acids crucial for compound activity are located at the interface between VSD and PD. Especially residues from the outer S5 helix orientated to VSD like Y267, L271 and G272 have a high impact on compound activity implicating possible interactions of (*R*)-L3 with the VSD as well (Fig. 2h). To analyze if the (*R*)-L3 interaction affects voltage-dependent movements of the VSD, we performed voltage-clamp fluoro-metry (VCF) experiments, that connect VSD movement to the

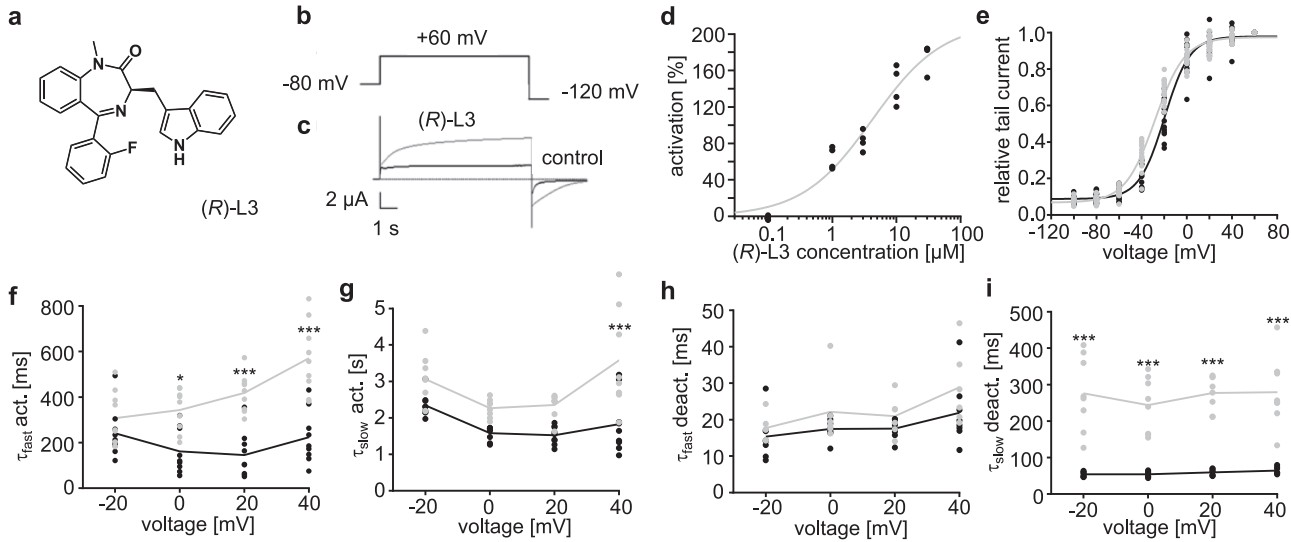

**Fig. 1 (R)-L3 activates and slows the rates of activation and deactivation of hKᵥ7.1 channels expressed in *Xenopus* oocytes. a** Chemical structure of (R)-L3. **b** Pulse sequences for voltage-clamp experiments. **c** Effect of 1 μM (R)-L3 on hKᵥ7.1 currents, recorded in an oocyte by a 7-s pulse to potentials of −100 mV to +60 mV from a holding potential of −80 mV. Currents were recorded in control solution containing 0.1% DMSO followed by perfusion with 1 μM (R)-L3 containing solution. **d** Dose–response curve for (R)-L3 from hKᵥ7.1 expressing oocytes at +40 mV test voltage. Each concentration was applied to four independent oocytes ($n = 20$) **e** Voltage dependence of current activation in the absence (black; $n = 13$) and presence of (R)-L3 (gray; $n = 15$) determined from peak tail currents measured at −120 mV. Currents were normalized to the peak tail currents elicited after a pulse to +40 mV. **f–i** Kinetics were evaluated in both absence (black; ctrl) and presence (gray; +(R)-L3) of 1 μM (R)-L3 and fitted by a two-component exponential function. Time constants are calculated for individual oocytes and given for fast (**f**; $n = 9$ for both conditions) and for slow (**g**; $n = 9$ for ctrl, $n = 8$ for + (R)-L3) component of Kᵥ7.1 activation as well as for fast (**h**; $n = 8$ for ctrl, $n = 6$ for + (R)-L3) and for slow (**i**; $n = 7$ for ctrl, $n = 8$ for + (R)-L3)) component of Kᵥ7.1 deactivation. Significance of mean differences was analyzed by one-way ANOVA and posthoc mean comparison Tukey test (*$p < 0.05$, ***$p < 0.001$).

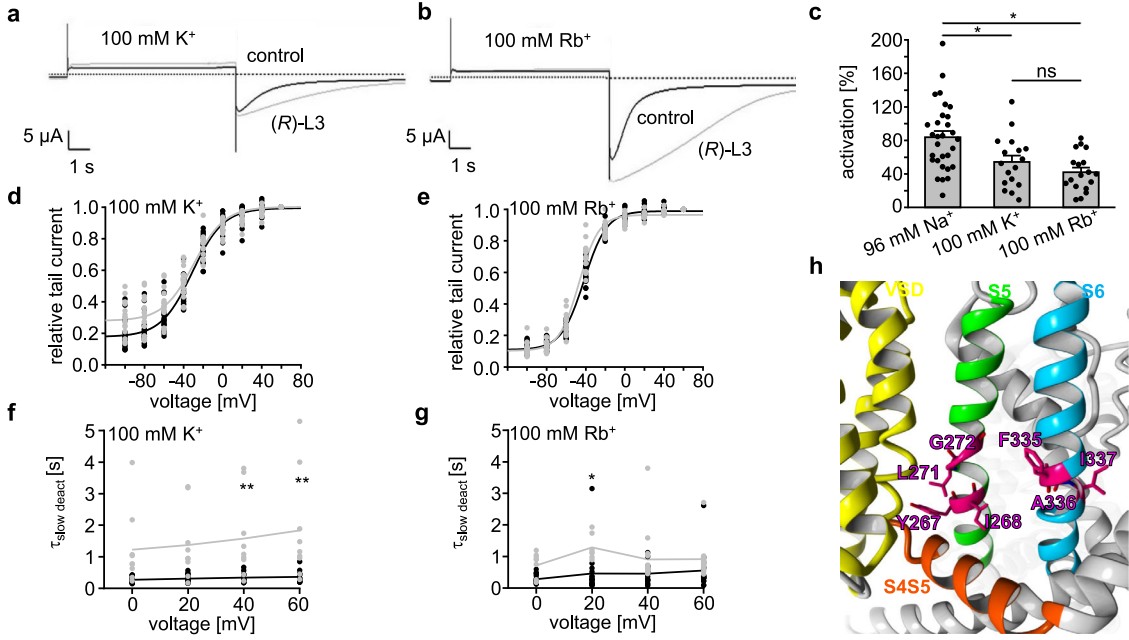

**Fig. 2 Effect of (R)-L3 on hKᵥ7.1 in high K⁺ and Rb⁺ buffer. a**, **b** Representative current traces in 100 mM K⁺ (**a**) and 100 mM Rb⁺ (**b**) before (black; ctrl) and after (gray; +(R)-L3) addition of 1 μM (R)-L3. **c** Activation of hKᵥ7.1 channel currents by 1 μM (R)-L3 expressed as percent change in current in high K⁺ and Rb⁺ ($n = 18$ for K⁺, $n = 17$ for Rb⁺) compared to activation measured in ND96 ($n = 30$). Significance of mean differences was evaluated by one-way ANOVA and posthoc mean comparison Tukey test (ns for $p > 0.05$, *$p < 0.05$). **d**, **e** Current-voltage relationship for hKᵥ7.1 expressing oocytes in the absence (black, $n = 18$ for K⁺ and Rb⁺) and presence (gray, $n = 17$ for K⁺, $n = 18$ for Rb⁺) of 1 μM (R)-L3 in 100 mM K⁺ (**d**) and 100 mM Rb⁺ (**e**). $V_{1/2}$ values were determined from normalized peak tail currents at −120 mV for each oocyte and fitted to Boltzmann equation. **f**, **g** Slow deactivation component of hKᵥ7.1 in high K⁺ (**f**) and high Rb⁺ (**g**) in the absence (black; $n = 15$ for K⁺, $n = 18$ for Rb⁺) and the presence (gray; $n = 9$ for K⁺ and $n = 18$ for Rb⁺) of (R)-L3. Time constants $\tau_{slow\ deact}$ were determined by two-component exponential fit for each oocyte and voltage step. **h** Depiction of KCNQ1 in activated state (AO, derived from Kuenze et al[38]. Amino acids crucial for (R)-L3 activity (magenta) are located at the lower S5 (green) and S6 (blue) helix. Most side chains of the crucial amino acids are orientated to the S4S5 Linker (orange) and VSD (yellow) of an adjacent KCNQ1 subunit.

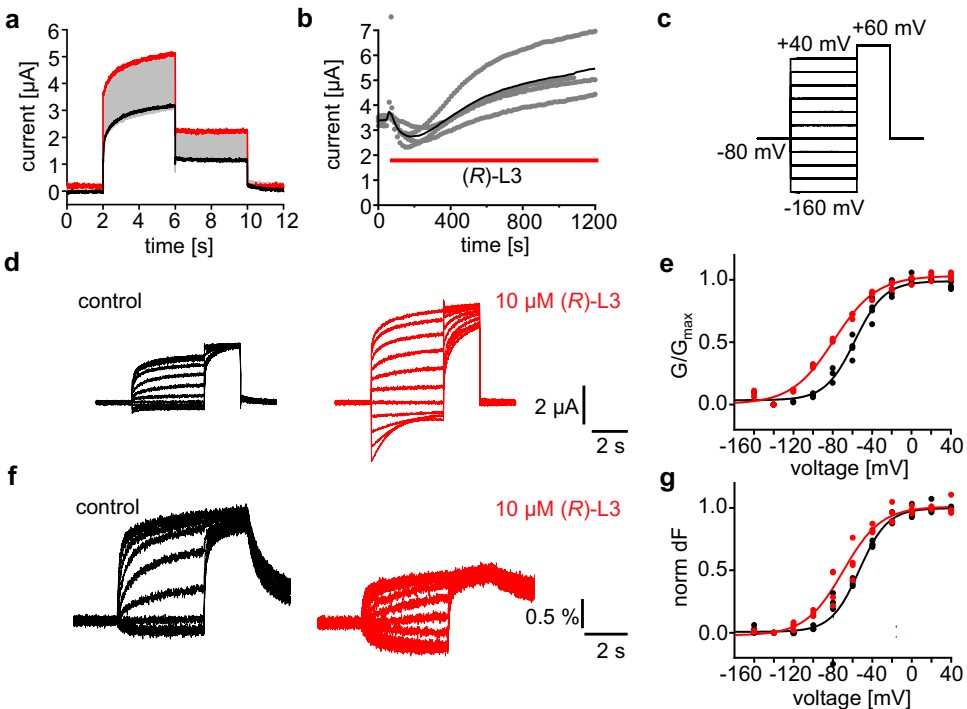

**Fig. 3 (*R*)-L3 potentiates K_v7.1_VCF (C214A/G219C/C331A) currents and left shifts the voltage-dependence of *G/G*_max and dF/F. a** Whole-cell currents from an oocyte expressing K_v7.1_VCF labeled with Alexa 488 C5 maleimide. Every 15 s, the membrane voltage was pulsed from the −80 mV resting potential to +60 mV for 4 s, followed by 2 s tails at −40 mV. Currents before (black) and after (red) a bolus of (*R*)-L3 was added to the bath (final concentration ~10 μM). **b** Steady-state current at +60 mV versus time during application of (*R*)-L3 (indicated by red bar). Blackline represents the mean value, gray dots represent raw data from each measurement (*n* = 4). **c** Pulse protocol for simultaneous current and fluorescence recordings in (**d**-**g**). **d** Sample current trace from a single oocyte before (black) and after (red) exposure to ~10 μM (*R*)-L3. **e** *G/G*_max voltage relationship of K_v7.1_VCF expressing oocytes in the absence (black) and presence (red) of (*R*)-L3 (*n* = 4 for both conditions). **f** Sample fluorescence trace from an oocyte before (black) and after (red) exposure to ~10 μM (*R*)-L3. **g** dF/F voltage relationship of K_v7.1_VCF expressing oocytes in the absence (black) and presence (red) of (*R*)-L3 (*n* = 4 for both conditions).

conducted current. We utilized the K_v7.1-construct K_v7.1_VCF in which two extracellularly accessible cysteines (C214A/C331A) were removed and a novel cysteine (G219C) was introduced in the VSD for site-specific attachment of Alexa 488-C5 maleimide. K_v7.1_VCF allows for monitoring of voltage-dependent VSD movements[30,31]. Since mutations can alter the activity of channel modulators, we first examined the effect of a high (*R*)-L3 concentration (10 μM) on the K_v7.1_VCF channel. As illustrated in Fig. 3a, b, 10 μM (*R*)-L3 are able to activate the K_v7.1_VCF channel by 58 ± 20% (*n* = 4, SEM). Although the activity of (*R*)-L3 is reduced compared to wildtype channels, the activation is still sufficient to analyze the effects of (*R*)-L3 on VSD movement by simultaneous recordings of current and fluorescence for each oocyte (Fig. 3c–g). Similar to the observed effects on the wildtype channel (*R*)-L3 shifts the $V_{1/2}$ of K_v7.1_VCF for *G/G*_max from −58 ± 2 mV (SE, *n* = 4) to −79 ± 3 mV (SE, *n* = 4) indicating a conserved mechanism of action for the mutated channel (Fig. 3d, e). To ensure that the shift of $V_{1/2}$ for *G/G*_max is not exclusively caused by the modulation of the pore domain, the change of fluorescence, which is caused by VSD movement, was simultaneously recorded. Detected fluorescence change (dF) clearly shows a similar left-shift of $V_{1/2}$ in the presence of (*R*)-L3 from −53 ± 1 mV (SE, *n* = 4) to −70 ± 1 mV (SE, *n* = 4) indicating an altered movement and therefore a direct effect on the VSD by (*R*)-L3 (Fig. 3f, g).

**Interactions with the lower S4 identified by alanine scanning.** Indeed, the large benzodiazepinic region of (*R*)-L3 was proposed to bind to the outer S5, which in turn is expected to interact with

the VSD of K_v channels in open or inactivated states[32]. Thus, the binding site may overlap with the interaction surface between VSD and PD. Specifically, the lower S4 helical transmembrane segment and the benzodiazepinic part may interact with similar regions of the lower S5. To identify single interacting amino acids, that are responsible for the modulation of VSD movements by (*R*)-L3, we performed alanine-scanning mutagenesis of the lower S4 transmembrane segment (residues 235–241) and evaluated if the modulatory effects of (*R*)-L3 on current amplitude and channel kinetics were preserved (Fig. 4). Since mutations in the S4 helix itself can alter channel properties, measurements of the same mutant or wildtype channel in the absence of (*R*)-L3 serve as a control (Fig. 4a–d). We found that (*R*)-L3 effects on current amplitude could be unaffected (I235A, M238A, L239A, and V241A), significantly increased (R237A and H240A) or decreased (L236A) by the respective amino acid exchange (Fig. 4e). These results indicate, that L236, R237, and H240 are involved in the modulatory mechanism of (*R*)-L3.

As the modulator slows activation and deactivation kinetics in wildtype channels the effect on kinetics of mutant channels was assessed and compared to the kinetics of the same mutant channel in the absence of (*R*)-L3. For all mutant channels except I235A no significant alteration of activation kinetics was observed indicating a direct or indirect involvement of these residues in the molecular mechanism of (*R*)-L3 (Fig. 4f, g). While the (*R*)-L3 effect of slowed deactivation is still present for L236A, H240A and V241A, the mutations I235A, R237A, M238A, and L239A show no impact on deactivation behavior in the presence of (*R*)-L3 (Fig. 4h, i). Interestingly, the previous coupled effects on

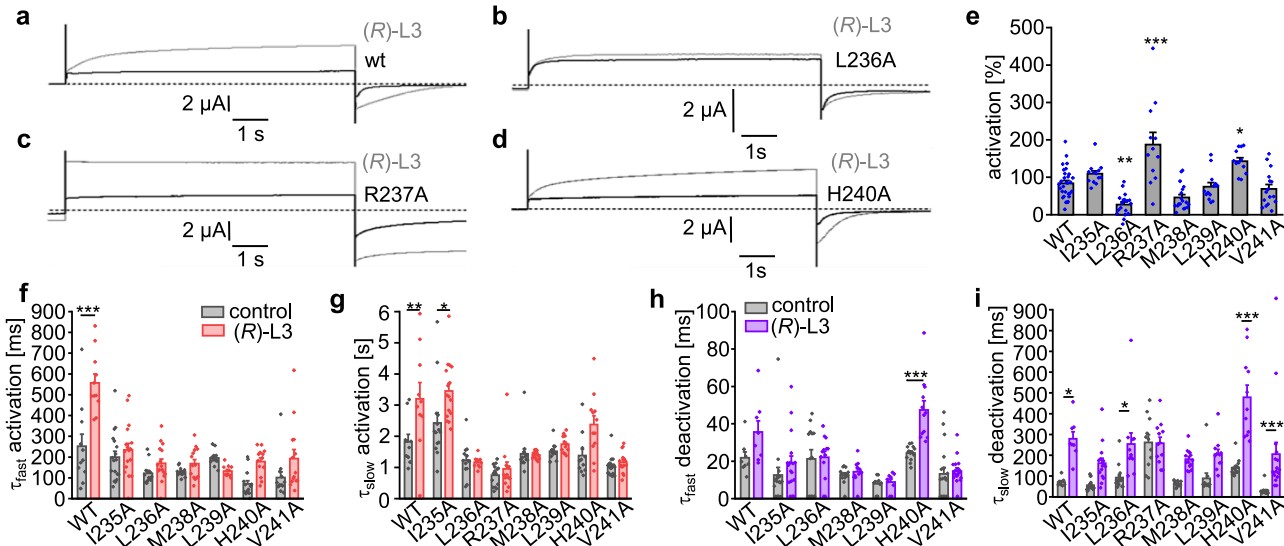

**Fig. 4 Sensitivity of S4 residues to (R)-L3 modulation. a–d** Representative current traces of wildtype (**a**) and mutant channels L236A (**b**), R237A (**c**) and H240 (**d**). **e** Current amplitude activation of $K_v7.1$ channel mutants by 1 μM (R)-L3 expressed as percent change in current measured at the end of a 7-s pulse to +40 mV ($n = 30$ for WT, $n = 13$ for I235A, $n = 18$ for L236A, $n = 12$ for R237A, $n = 18$ for M238A, $n = 13$ for L239A, $n = 12$ for H240A, $n = 16$ for V241A). Significance of mean differences was evaluated by one-way ANOVA and posthoc mean comparison Tukey Test (*$p < 0.05$, **$p < 0.01$, ***$p < 0.001$). **f–i** Effect of (R)-L3 on h$K_v7.1$ channel mutant activation and deactivation rates. Kinetics were evaluated in both absence and presence of 1 μM (R)-L3 and fitted by two-component exponential function (number of independent oocytes for (**f-i**) are given in SI Table 3).

channel kinetic and current amplitude by (R)-L3 become more uncoupled by the examined mutations of the lower S4. Summarizing the results, only mutations from R237 to L239 were able to eliminate both effects on channel activation and deactivation kinetics. Out of these residues, the mutation M238A showed the most prominent reduction of the agonistic effect on the current amplitude.

**M238 plays a crucial role in (R)-L3 mechanism and VSD-PD-coupling.** To assess the relevance of key interacting residue M238 for (R)-L3 agonism in normal channel gating we conducted photo-crosslinking experiments (Fig. 5a–f). Modeling suggests a possible physical interaction of M238 with outer pore of a neighboring subunit by the S5 residue I274 in the activated state (Fig. 5c). This hypothesis is supported by the published cryo-EM structures of KCNQ1 (PDB ID 6UZZ), which shows the VSD in an activated state[33]. The rearrangement of the VSD leads to distinct rotation as well as upward movement of S4 resulting in direct interaction with S5[34]. Consequently, the distance between M238 and I274 is reduced *in silico* entering a range, where direct interactions are possible. The putative interaction was evaluated by incorporating the photo-activatable non-canonical amino acid *p*-azido-phenylalanine (AzF) at position $K_v7.1$ M238 (M238AzF) using the amber suppression method in HEK cells (Fig. 5a–d)[35]. Application of high $K^+$ buffer (137 mM) depolarized cells favoring voltage sensors in the activated up-state, which led to M238AzF-mediated inter-subunit photo-crosslinking, as shown by the formation of $K_v7.1$ multimers in Western blots (Fig. 5e). These observations are consistent with the proposed interaction between M238AzF and a neighboring subunit pore domain (S5-S6), which is expected to result in covalent inter-subunit bonds. As an alternative but much less site-specific approach, we assessed the global incorporation of photo-methionine (pMet). Similar to the results for M238AzF, pMet incorporation allowed for photo-crosslinking and stable formation of $K_v7.1$ multimers in Western blots in high $K^+$ solution (Fig. 5e). These two different crosslink experiments support a M238-inter-subunit interaction. To verify this again, we chose a second experimental approach

addressing the inter-domain interaction with $Co^{2+}$ ions, that can facilitate cysteine bridges, and the $K_v7.1$ double mutant M238C/I274C (see materials and methods). Oocytes expressing the wildtype channel as well as the single mutants $K_v7.1$ M238C and $K_v7.1$ I274C serve as negative controls (Fig. 5f, SI Fig. 2). For $K_v7.1$ wt, $K_v7.1$ M238C and $K_v7.1$ I274C expressing oocytes no significant changes of activation currents were observed in the presence of $Co^{2+}$. However, the double mutant $K_v7.1$ M238C/I274C showed significantly reduced activation current in the presence of $Co^{2+}$ ions indicating locked VSD and PD interaction. Interestingly, the reduction of activation current was reversible by the addition of (R)-L3. Together with the results from western blotting these data clearly show the importance of M238 and I274 interaction for channel opening. The data extend and support the previous findings of the coupling mechanism and characterize M238 as a key residue for VSD coupling with the pore[36,37].

**In silico analyses of the (R)-L3 binding process to the $K_v7.1$ channel.** To connect the experimental data with structural analyses of the ligand/receptor complex and the protein movement, we used previously described models of different ion channel states for docking of (R)-L3 and subsequent molecular dynamics (MD) simulations[38]. The used ion channel models are homotetrameric structures composed of four α-subunits (Mol A–Mol D) and display the receptor with an activated VSD and an open pore (AO), an activated VSD and a closed pore (AC) and a resting VSD with closed pore (RC). Using the results from mutational analysis we built complexes of (R)-L3 and the three different receptor states (AO, AC, RC) by manually docking the ligand into the pocket between S4 and the S4S5 linker of Mol B with the indol-3-yl moiety orientated between the α-helices S5 and S6 from Mol A (Fig. 6a–c). The generated ligand/receptor complexes were energy minimized and embedded into lipid membranes surrounded by water molecules (Fig. 6d). After second annealing and energy-minimization step, MD simulations with a duration of 30 ns were performed 3 (RC, AC) or 5 (AO) times for the generated ligand/receptor complexes.

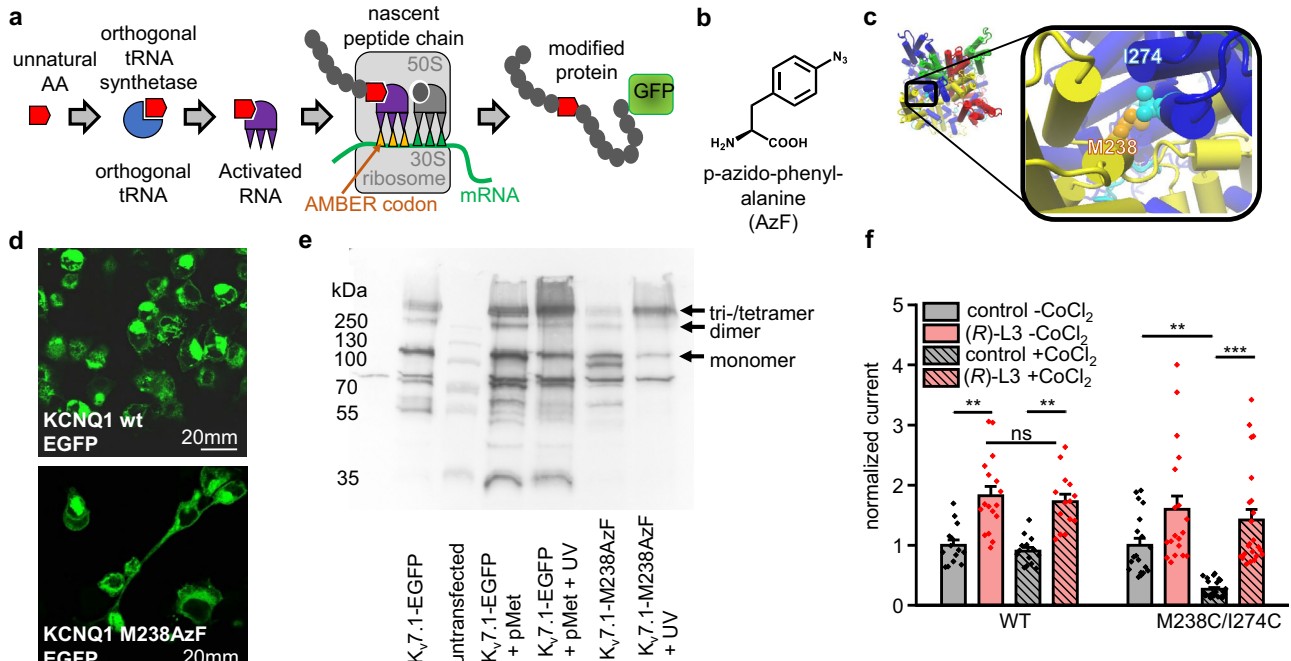

**Fig. 5 Photo-crosslinking of the photo-activatable non-canonical amino acid AzF leads to formation of multimers. a, b** Using the amber suppression methodology for incorporation of non-canonical amino acids (ncAAs), we introduced the photo-activatable non-canonical amino acid AzF (**b**) at M238 position into mutant $K_v7.1$-EGFP-M238AMBER-Stop. HEK cells were co-transfected with cDNA encoding $K_v7.1$-EGFP-M238amber-Stop, suppressor tRNA and AzF-tRNA synthetase. Addition of the ncAA p-azido-phenylalanine (AzF) at 0.5 mM to the cell culture medium allowed for incorporation of the ncAA. **c** I274 interacts with M238 from the adjacent subunit in activated state *in silico*. **d** Successful incorporation and full-length protein expression was assayed by confocal EGFP imaging. **e** $K_v7.1$-EGFP-M238AzF expressing HEK cells were incubated in high $K^+$ solutions (137 mM KCl) leading to channels preferentially in depolarized (calculated $V_m$ of about $-8.9$ mV) states. UV irradiation caused formation of multimers under high $K^+$ / preferentially depolarizing conditions consistent with the predicted interaction. **f** Normalized currents of $hK_v7.1$ WT and $hK_v7.1$ M238C/I274C expressing oocytes in the absence and presence of (*R*)-L3 and $Co^{2+}$ ions (number of oocytes for each condition see SI Table 2).

For analysis of MD simulations, we used parameters like the root mean square deviation (RMSD), root mean square fluctuation (RMSF) and the movement correlation of residues expressed by dynamic cross-correlation matrices (DCCMs). The RMSD can be calculated for multiple or single amino acids or even atoms and displays the total movement from the beginning of the simulation to the end, while RMSF displays the fluctuation around a mean position. After 30 ns of simulation, the total RMSD of the ligand/receptor complexes was constantly fluctuating around a steady-state value indicating a stable formation. Although the total RMSDs decrease from the AO-complexes to the RC-complexes, the means were not significantly different (Fig. 6e). In contrast, the RMSD for (*R*)-L3, which displays the movement of the ligand away from the energy minimized starting position, was significantly reduced for the AO-complex (Fig. 6f). Since the RMSF of the ligand did not significantly differ, the different RMSDs must be caused by a movement of the complete ligand instead of a reorientation of ligand site chains (Fig. 6g). All simulated complexes show a constantly fluctuating ligand RMSD after an equilibration time of 10 ns. Therefore, the binding energy for every ligand/receptor complex was calculated from 10 ns to 30 ns every 0.25 ns. The highest binding energies were observed in the AO-complexes indicating a better binding of (*R*)-L3 to this ion channel state (Fig. 6h). Together with the reduced ligand RMSD in the AO state, these results lead to the conclusion of a favored binding to the fully activated AO receptor state.

**Aromatic interactions between (*R*)-L3 and W248 are crucial for the activity.** Hence, we focused on the AO-ligand/receptor complex and analyzed interactions between (*R*)-L3 and the

mutated amino acids of the lower S4 segment. Only hydrophobic interactions with L236, M238, L239, and V241 were detected. Additionally, a strong aromatic interaction between W248 from the S4S5 linker (Mol B) and the 2-chlorophenyl moiety was observed. Figure 6i shows the relative interaction duration over the complete simulation time, which indicates a strong influence of W248 on the binding. Consequently, we included W248 into further analyses. While distinct interactions with the receptor are needed for compound binding, the mechanism of activation is conducted by altered movements of the protein in the presence of the ligand. To detect these movements, we performed three additional 30 ns MD simulations of the AO state model without the ligand for comparison. Surprisingly, the examined amino acids in S4 (Mol B) did not show significantly different RMSD or RMSF values indicating a similar behavior for the presence and absence of (*R*)-L3 (Fig. 6j, k). In contrast, the RMSD of W248 from the S4S5 linker (Mol B) was significantly reduced without a significant change in the RMSF suggesting an inhibited movement of the amino acid in the presence of (*R*)-L3.

Due to the frequently detected aromatic interactions with the ligand *in silico* and the altered behavior of W248 in the presence of (*R*)-L3 additional mutants W248F, W248A, and W248R were generated. TEVC measurements revealed a significant loss of (*R*)-L3 activity for all mutants compared to wildtype emphasizing the involvement of W248 in the molecular mechanism of action (Fig. 6l). Comparing the activity of 1 μM (*R*)-L3 on aromatic mutant W248F ($34 \pm 6\%$, SEM, $n = 5$) with activity on W248A ($20 \pm 6\%$, SEM, $n = 5$) and W248R ($10 \pm 3\%$, SEM, $n = 4$) lead to the conclusion that not only the size but also the aromaticity of the amino acid at this position seems important for the agonistic

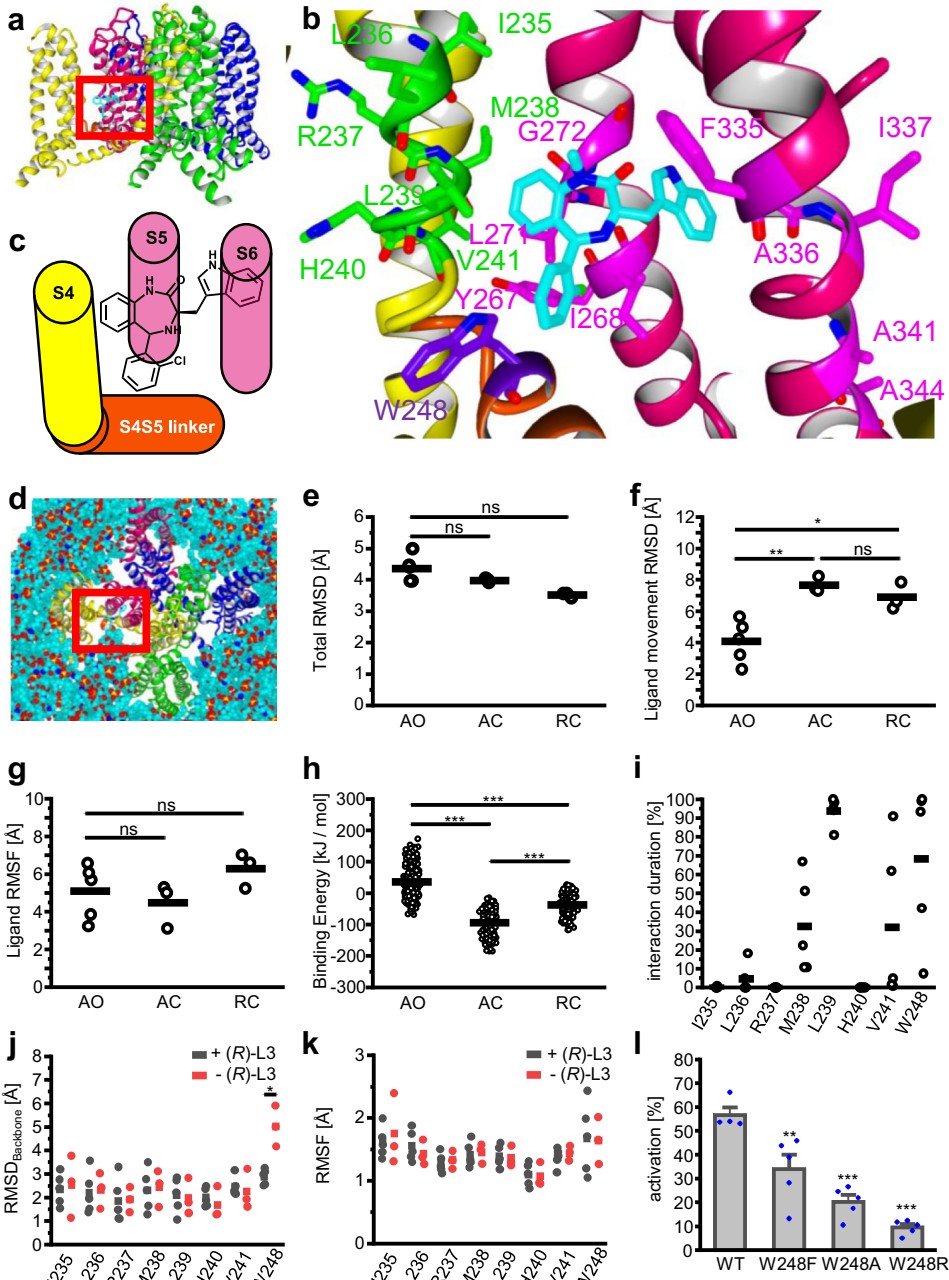

**Fig. 6 Binding site localization and *in silico* simulation results. a** Depiction of tetrameric KCNQ1 AO model with subunits colored in magenta (Mol A), yellow (Mol B), green (Mol C) and blue (Mol D). Proposed binding site of (*R*)-L3 is marked by red square. **b** Close-up depiction of (*R*)-L3 (CPK-color coded with cyan for carbon) binding site with S4 (yellow) and S4S5 linker (orange) from subunit Mol B and S5 and S6 helix (magenta) from subunit Mol A. Previous analyzed amino acids from S5 and S6 with impact on (*R*)-L3 activity are colored in purple, while newly analyzed residues from S4 and S4S5 linker are colored in green or violet. **c** Schematic illustration of (*R*)-L3 binding site between S4 (yellow) and S4S5 linker (orange) from subunit Mol B and S5 and S6 (pink) from subunit Mol A. **d** Depiction of KCNQ1 AO model embedded in membrane for MD simulation. Subunits Mol A–D are colored like in 5A and (*R*)-L3 binding site is marked by red square. **e** Root mean square deviation (RMSD) of 30 ns MD simulations for complete modeled structures AO (n = 5 simulations), AC (n = 3) and RC (n = 3) with (*R*)-L3 for every simulation. Means are shown as bar and have no significant differences indicated by ns. **f** RMSD of Ligand movement of (*R*)-L3 from start to end of the 30 ns MD simulation of KCNQ states AO (n = 5), AC (n = 3) and RC (n = 3). **g** (*R*)-L3 Root mean square fluctuation (RMSF) for 30 ns MD simulation with KCNQ1 state models AO (n = 5), AC (n = 3) and RC (n = 3). Mean differences are not significant (ns; p > 0.05). **h** Binding Energy [kJ/mol] of (*R*)-L3 calculated for all simulation snapshots from 10 to 30 ns of MD simulations for KCNQ1 models AO (n = 405 snapshots), AC (n = 243) and RC (n = 243). **i** Percentage duration of hydrophobic interaction between (*R*)-L3 and mutated amino acids from S4 and S4S5 linker over total MD simulation time of 30 ns for AO simulations (n = 5). **j, k** Backbone RMSD and RMSF of mutated residues in the absence (n = 3 simulations)/presence (n = 5 simulations) of (*R*)-L3. **l** activity of 1 μM (*R*)-L3 at wildtype KCNQ1 expressing oocytes compared to mutations W248F, W248A, and W248R (n = 4 for WT, n = 5 for W248F, W248A, and W248R).

mechanism. Since aromatic interactions are also possible with phenylalanine the mutant W248F represents only a partial disruption of compound/channel interaction at this position. This hypothesis is also consistent with the activation and deactivation parameters of the W248 mutants in the presence and absence of (R)-L3 (SI Fig. 3). Whereas no significant alteration of activation or deactivation time constants upon (R)-L3 application is observed at W248A and W248R expressing oocytes representing a disruption of kinetic modulating effects by (R)-L3, W248F expressing oocytes show significant alteration in deactivation kinetics in the presence of (R)-L3, which is also found in wildtype channels.

**VCF measurements for putative interacting amino acids**. To characterize the influence of the most important interacting amino acids M238, V241, and W248 in more detail we tried to record further VCF experiments using the alanine mutants of these residues. However, sufficient fluorescence signals were only detected for the mutation V241A (SI Fig. 4). The fact that low currents in TEVC mirrors insufficient labeling site-specific attachment of Alexa 488-C5 maleimide allowing for VSD suggests weak plasma membrane expression of these mutants as described for S4 mutated KCNQ1 channels before[39–41]. As VCF is a method with low signal-to-noise ratio, several mutated weakly expressing $K_v7.1$ channels could not be analyzed. Compared to the wildtype channel, $K_v7.1_{VCF}$ V241A showed right-shifted $V_{1/2}$ values of $-29 \pm 1$ mV ($G/G_{max}$) and $-25 \pm 1$ mV (dF/F). In the presence of (R)-L3 the typical left-shift to more negative values is observable with $V_{1/2}$ values of $-35 \pm 1$ mV ($G/G_{max}$) and $-29 \pm 2$ mV (dF/F). However, compared to the wildtype these shifts are much less pronounced consistent with a direct involvement of compound/residue interaction in the agonistic mechanism.

**Influence of (R)-L3 binding on VSD-PD-coupling *in silico***. While only minor differences are found on the scale of single interacting amino acids, major differences were observed in the communicational behavior of the binding site forming segments S4 (Mol B), S4-S5 linker (Mol B), S5 (Mol A), and S6 (Mol A) (Fig. 7a). To visualize the altered communication between the segments, DCCMs were calculated for all simulations and averaged depending on the presence or absence of (R)-L3 (Fig. 7b–j). DCCM was calculated for a pair of Cα atoms over the complete MD simulation time and displayed movement behavior of the selected atoms in a range from $-1$ (fully anticorrelated movements) over 0 (no movement correlation) up to 1 (fully correlated movements). Although the S4 helix (Mol B) and the S4S5 linker (Mol B) are in close distance and on the same subunit their movements were not coordinated in the simulations without (R)-L3 (Fig. 7b). In the presence of the ligand a clear increase of coordinated movements could be observed between the lower S4 segment with the adjacent residues from the S4S5 linker (Fig. 7c, d). In contrast, the natural communication and the movement correlation between S4 (Mol B) and the corresponding S5 helix (Mol A) was nearly abolished (Fig. 7e–g). In accordance with the reduced correlation between S4 and S5 also the simultaneous movements of S5 and the S4S5 linker were reduced (SI Fig. 5a–c). The presence of (R)-L3 generated also weak coupling between S4 (Mol B) and S6 (Mol A), while under normal condition no long-range coupling could be detected between these segments (Fig. 7h–j). In particular, the movement correlation between M238 and surrounding residues from S4 with the residues A336 – F340, which are directly interacting with (R)-L3, was increased. However, these differences were less pronounced than the coupling changes between S4, S4S5 linker and S5. A more

pronounced elevation of coupling was achieved by (R)-L3 for the lower S6 helix with the whole S5 helix (SI Fig. 5d–f), while at the same time S6 was uncoupled from the movements of the S4S5 linker (SI Fig. 5g–i). Summarizing the movement coupling, the presence of (R)-L3 separated the movements of the VSD from the PD by decreasing the correlation between S4 and S5 and simultaneously increasing the correlation between S4 and S4S5 linker and between the inner pore domain forming helices S5 and S6.

Since (R)-L3 increases $K_v7.1$ current amplitude, an allosteric coupling between the binding site forming structures and the adjacent selectivity filter residues can be assumed (Fig. 8a). Hence, RMSDs of amino acids V310–P320 (Mol A) and the coupling of these with S4, S4S5 linker, S5 and S6 were analyzed (Fig. 8b–o). Comparing the residue backbone RMSD in the absence and presence of (R)-L3, the amino acids V310–T312 and K318–P320 showed a moderate, but not significant increased movement when the ligand is bound (Fig. 8b). However, the averaged backbone RMSD for V310–P320 was significantly increased suggesting an elevated movement of the whole selectivity filter segment (Fig. 8c). Similar to previous results for S4, the movement correlation of S4 with the selectivity filter decreased. In particular, the correlation between M238 and the lower residues was extinguished by the ligand (Fig. 8d–f). In contrast, a slightly increased movement correlation occurred between the S4S5 linker and the lower amino acids of the selectivity filter (Fig. 8g–i). While the outer structures showed only minor correlation differences with V310–P320, the two inner helices S5 and S6 nearly swap their behavior towards the selectivity filter (Fig. 8j–o). S5 loosed correlation especially to the amino acids T312–K318, while S6 achieves a higher grade of correlation with the lower selectivity filter. Especially F339, which is directly connected to the indol-3-yl moiety of (R)-L3 via aromatic interactions, amplified its movement coupling with V310–I313. In summary, (R)-L3 raised the mobility of V310–P320 and altered the coupling of these residues by enhancing the movement correlation with S6 and simultaneously decreasing it with S4 and S5.

## Discussion

In the present study, we analyzed the binding site and mechanism of action of the known $K_v7.1$ modulator (R)-L3, which slows down channel activation and deactivation and increases current amplitude (Fig. 1)[23]. Current amplitude, inactivation, activation and deactivation kinetics are dependent on the external ion species, via allosteric effects within the pore domain[19,26]. Based on this phenomenon, we used different external ion solutions to evaluate if the effects of (R)-L3 are resulting from direct modulation of the PD. Indeed, the increment of corrected tail current amplitude by (R)-L3 is reduced in 100 mM $K^+$ as well as in 100 mM $Rb^+$ and therefore dependent on the external ion species and concentration (Fig. 2c). On the other hand, residues in the S5-S6 pore domain have been identified to be involved in both, the formation of the (R)-L3 binding site[23] as well as the formation of an allosterically linked cluster among lower selectivity filter. In particular, Y267, I268, and L271 are crucial for the activity of (R)-L3. These residues were also directly involved in the *in silico* binding of (R)-L3 forming aromatic and hydrophobic interactions with the compound (Fig. 6b). Thus, S5 and S6 determine current amplitudes and inactivation[19,26,42]. The modulator may allosterically influence the conduction pathway to allow for increased conduction rates. This is also indicated by the increased RMSD of the residues forming the selectivity filter (Fig. 8c). In contrast, (R)-L3 effects on current deactivation kinetics are similar in high $Na^+$, $K^+$ and $Rb^+$ (Fig. 2F, G, SI Fig. 1) which indicates that the activity of the modulator can be functionally

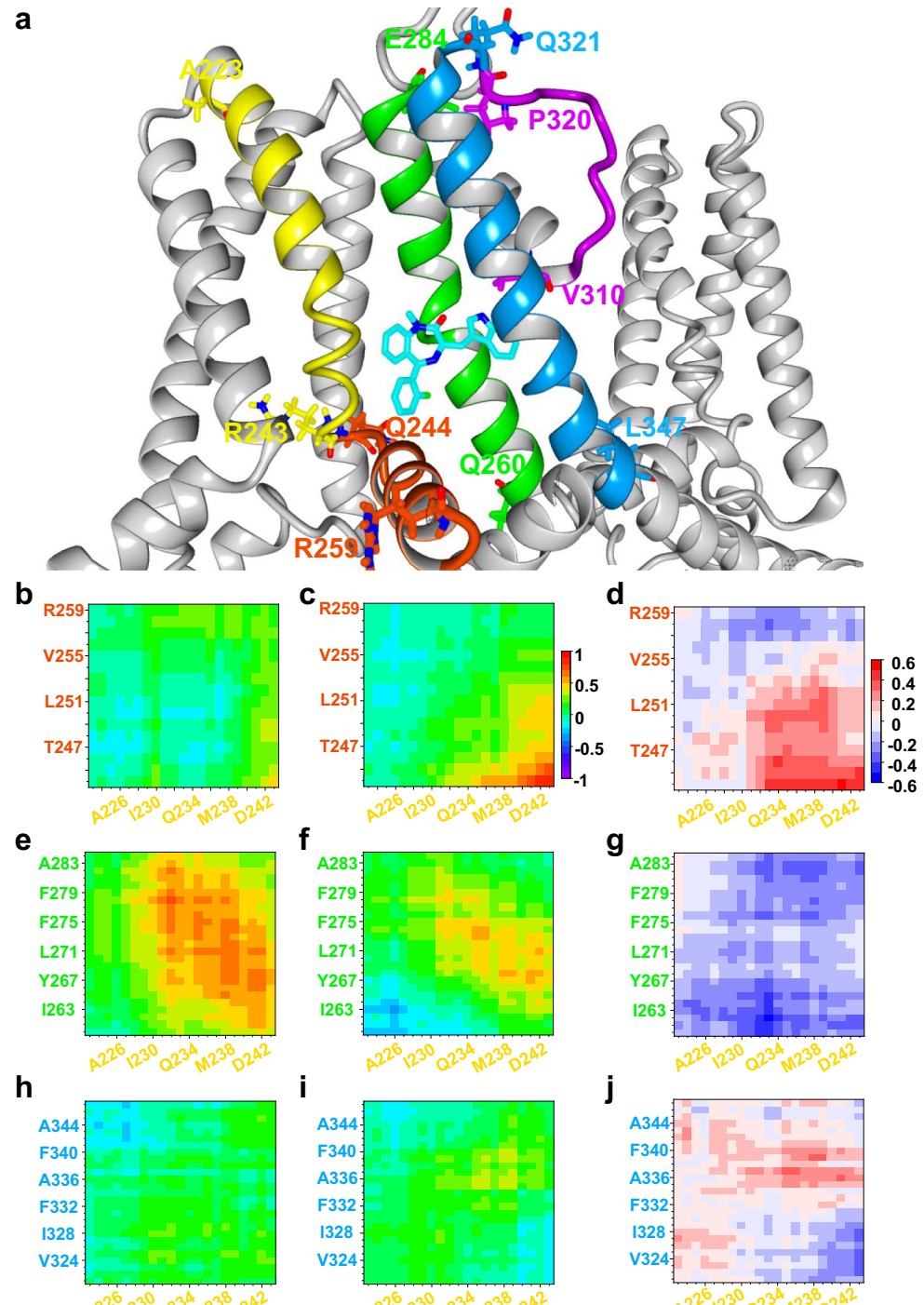

**Fig. 7 Dynamic cross-correlation matrices (DCCMs) for interactions between VSD (S4, S4S5 linker) and PD (S5, S6). a** (*R*)-L3 (CPK-color code with cyan for carbon) binding site between subunit Mol A and B with interacting secondary structures S4 (Mol B, yellow), S4S5 linker (Mol B, orange), S5 (Mol A, green), S6 (Mol A, blue) and selectivity filter (Mol A, purple). First and last amino acid of all secondary structures are shown in the same color. **b**, **c** Dynamic cross-correlation matrix (DCCM) for S4 (Mol B, yellow) and S4S5 linker (Mol B, orange) in the absence (**b**) and presence (**c**) of (*R*)-L3 from −1 (fully anticorrelated) over 0 (not correlated) to 1 (fully correlated). **d** Increase (positive values, red) and decrease (negative values, blue) of correlation between S4 and S4S5 linker residues depending on the presence of (*R*)-L3. **e**, **f** DCCM for S4 (Mol B, yellow) and S5 (Mol A, green) in the absence (**e**)/ presence (**f**) of (*R*)-L3. **g** Increase (positive values, red) and decrease (negative values, blue) of correlation between S4 and S5 residues depending on the presence of (*R*)-L3. **h**, **i** DCCM for S4 (Mol B, yellow) and S6 (Mol A, blue) in the absence (**h**)/presence (**i**) of (*R*)-L3. **j** Increase (positive values, red) and decrease (negative values, blue) of correlation between S4 and S6 residues depending on the presence of (*R*)-L3. All data of Fig. 7 are derived from 5 independent simulations in the presence and 3 independent simulations in the absence of (*R*)-L3.

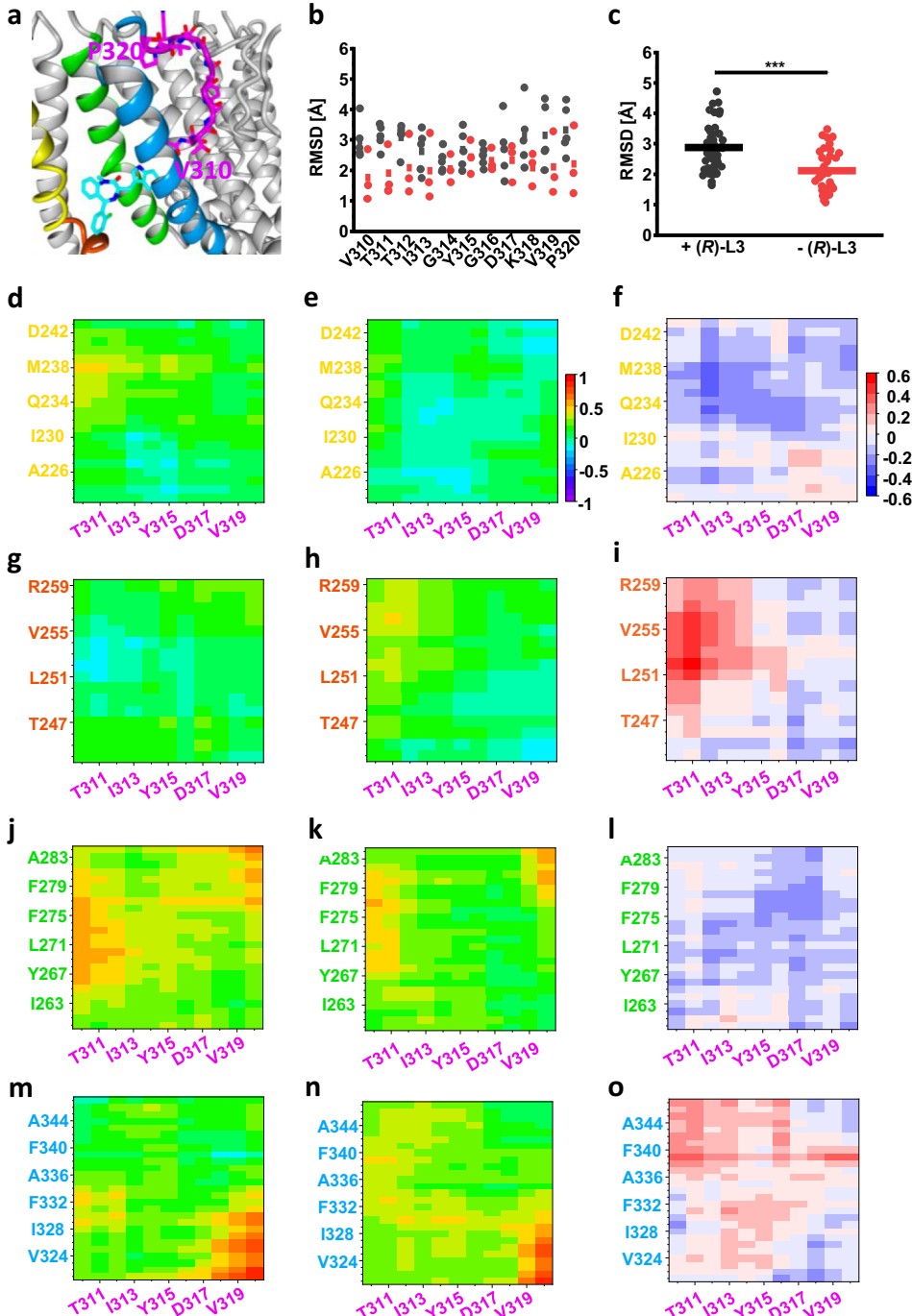

**Fig. 8 Dynamic cross-correlation matrices (DCCMs) for interactions between Pore helix and S4-S6. a** Close-up depiction of (R)-L3 binding site with S4 (yellow), S4S5 linker (orange), S5 (green), S6 (blue) and selectivity filter (purple) with all amino acids from V310 to P320. **b, c** RMSD of selectivity Filter residues V310 – P320 individually (**b**) and in total (**c**) in the absence (black) and presence (red) of (R)-L3. Means are given as squares (**b**) or bars (**c**). (**d, e**) Dynamic cross-correlation matrix (DCCM) for S4 (Mol B, yellow) and selectivity filter (Mol A, purple) in the absence (**d**) and presence (**e**) of (R)-L3 from −1 (fully anticorrelated) over 0 (not correlated) to 1 (fully correlated). **f** Increase (positive values, red) and decrease (negative values, blue) of correlation between S4 and selectivity filter residues depending on the presence of (R)-L3. **g, h** DCCM for S4S5 linker (Mol B, orange) and selectivity filter (Mol A, purple) in the absence (**g**)/presence (**h**) of (R)-L3. (**i**) Increase (positive values, red) and decrease (negative values, blue) of correlation between S4S5 linker and selectivity filter residues depending on the presence of (R)-L3. **j, k** DCCM for S5 (Mol A, green) and selectivity filter (Mol A, purple) in the absence (**j**)/presence (**k**) of (R)-L3. **l** Increase (positive values, red) and decrease (negative values, blue) of correlation between S5 and selectivity filter residues depending on the presence of (R)-L3. **m, n** DCCM for S6 (Mol A, blue) and selectivity filter (Mol A, purple) in the absence (**m**)/presence (**n**) of (R)-L3. **o** Increase (positive values, red) and decrease (negative values, blue) of correlation between S6 and selectivity filter residues depending on the presence of (R)-L3. All data of Fig. 8 are derived from 5 independent simulations in the presence and 3 independent simulations in the absence of (R)-L3.

separated in a stimulation effect of current amplitude via modulation of the conduction pathway and a second (*R*)-L3 effect on current kinetics by a different mechanism.

The (*R*)-L3 impact on current kinetics could result from altered voltage-sensor movement. To confirm this hypothesis, voltage-clamp fluorometry (VCF) was performed and showed that the (*R*)-L3 effects are followed by parallel left shifted GV and FV curves (Fig. 3). This finding is consistent with mathematical modeling suggesting that the modulator favors opened over closed states[23]. Thus, beside direct effects on the pore domain separate effects on the VSD are observable.

By inspection of the $K_v7.1$ homology model the identified S5-S6 residues suggest a binding site from the interface of S5 and S6 (indol-3-yl sidechain) to the outer S5 (benzodiazepinic part). Especially in activated VSD state, the outer S5 of $K_v$ channel subunit is expected to be located in close proximity of the primary voltage sensing helix S4 of the adjacent subunit, suggesting that (*R*)-L3 may interact with S4 as well. Usage of classical mutagenesis scanning of the lower S4 identifies several residues modulating the activation of $K_v7.1$ channels. The most dramatic impacts on (*R*)-L3 effects were seen with mutation at R237 and M238 whose significantly blunted current potentiation and kinetic modulation (Fig. 4, SI Fig. 3a). Other mutants impaired compound effects on either activation or on deactivation or on current amplitudes but not on all parameters. However, due to sidechain orientation direct interactions between residue and compound are only possible for M238 *in silico* inspecting both available cryo-EM structures[33,43]. Therefore, M238 may be considered a key residue for (*R*)-L3 activity.

Recently, a two-stage hand-and-elbow gating mechanism for $K_v7.1$ was proposed. S4 movement towards a fully activated conductive state depends on an elbow-like hinge between S4 and S4S5 linker that engages with the pore of the neighboring subunit to activate conductance[37]. Based on this open state model interaction of M238 with the neighboring S5-S6-domain can be predicted *in silico*. In order to test this hypothesis in vitro we adapted a photo-crosslinking approach and show crosslinking of M238AzF with the neighboring subunit in the open state (Fig. 5). Further, the functional influence of the inter-domain interaction of M238 was assessed by $CoCl_2$ confirming our hypothesis. Since the distance between M238 and I274 is depending on VSD conformation, the interaction between these residues is favored in the activated VSD state. Thus, M238 was identified as a key residue for (*R*)-L3 activity as well as a potential key player for the VSD-PD-coupling.

In order to evaluate the binding site further we used different $K_v7.1$ channel models (AO, AC, and RC state) and manually docked the (*R*)-L3 molecule assuming close proximity of key residues. Thus, guided by the experimentally identified key residues in S5/S6[23] and S4 (M238) we developed an *in silico* model (Fig. 6). In MD simulations (*R*)-L3 was stably coordinated over $24 \pm 5.45$ ns ($n = 5$) only to the AO model to the postulated binding pocket in close proximity of the key residues suggesting favorable binding only in the fully activated state. By inspection of the model complex, residue W248 in the S4S5 linker was identified as a likely interaction partner. To verify our model and the influence of W248 in vitro, we mutated the natural tryptophan and found that it is highly important for the activity of the modulator on the $K_v7.1$ channel (Fig. 6d). Therefore, extensive evidence suggests that (*R*)-L3 binds in a binding pocket formed by S4, S4-S5 linker and the pore domain S5/S6 of the neighboring $K_v7.1$ channel subunit, which is exactly the key region of electromechanical coupling in $K_v7.1$-voltage-dependent activation[37]. (*R*)-L3-induced uncoupling of key residue M238 from the neighboring S5-S6-domain and the lower $K^+$-coordinating residue T331 in the lower selectivity filter in the open state may allow

for higher conduction. Stabilization of interactions between the hand-like C-terminus of the VSD-pore linker (S4-S5L) with the pore (S5-S6) stabilizes the fully activated state conformation as suggested by kinetic Markov modeling. Further, a straightened S6 helix allows for KCNQ1-typical structure-function coupling with impact on deactivation and selectivity filter function[42,44]. In addition, this hypothesis is in good accordance with the correlated movement of M238 and the segments S4S5 linker and selectivity filter. Without (*R*)-L3 the movement of M238 correlates with residues from the selectivity filter (Fig. 8d; residues V310-Y315), especially with T312, while no coordinated movement can be seen with the S4-S5 linker (Fig. 7c). In the presence of (*R*)-L3 this behavior towards the linker and the selectivity filter is reversed underlining the separation of VSD and pore domain by the compound.

Recently, quercetin was identified as activator of homomeric KCNQ1 channels[45]. The mechanism of KCNQ1 current potentiation results from an accelerated activation as well as decelerated deactivation. Similar to (*R*)-L3, a possible binding site for the compound was detected in close proximity to F340, which is located on the lower S6 helix. Functional analysis of F340V show, that this residue is directly or allosterically involved in the activation mechanism provided by quercetin. On the other hand, docking studies as well as further functional analyses indicate quercetin binding to at least one additional and independent binding site, which is also functionally important for the mechanism of action. Thus, quercetin seems less specific and less potent compared to (*R*)-L3.

Summarizing, this study provides evidence for two individual molecular mechanisms allowing $K_v7.1$ channel activation by (*R*)-L3: Firstly, current amplitudes are increased due to compound binding to the pore domain allosterically modulating the ion pathway. Secondly, $K_v7.1$ channels may be trapped in open states by the stabilization of the voltage sensors in the activated state (up-state).

## Materials and methods

**Mutagenesis and heterologous expression of potassium channels**. Site-directed mutagenesis of $K_v7.1$ cDNA (GenBank Acc. No. NM_000218) subcloned into the pXOOM vector was performed using standard mutated oligo extension PCR. Mutations were verified by automated DNA sequencing. The constructs were linearized with NheI, and in vitro transcriptions were performed using the mMessage mMachine kit (Ambion) with T7 polymerase according to the manufacturer's instructions. The quality of synthesized RNA was checked by gel electrophoresis and quantified by spectrophotometry.

Oocytes were isolated from ovarian lobes of *Xenopus laevis* and enzymatically digested with collagenase (Type II, Worthington, 1 mg/ml in calcium-free Barth's solution) for about 1.5–2 h, as previously described. Stage IV or V oocytes were injected with 5 ng cRNA encoding either wildtype or mutant $K_v7.1$ subunits and stored for 3–4 days at 18 °C in Barth's solution containing (in mmol $L^{-1}$): 88 NaCl, 1.1 KCl, 2.4 NaHCO₃, 0.3 Ca(NO₃)₂, 0.3 CaCl₂, 0.8 MgSO₄, 15 HEPES-NaOH, penicillin-G (31 mg $L^{-1}$), gentamycin (50 mg $L^{-1}$), streptomycin sulfate (20 mg $L^{-1}$), pH 7.6 before electrophysiological recording.

**Electrophysiology and $Co^{2+}$ crosslinking**. Whole-cell currents in oocytes were recorded at room temperature by two-electrode voltage-clamp (TEVC) using a Turbo Tec 10CD amplifier (NPI Electronic GmbH, Tamm, Germany) and an ITC-16 interface combined with GePulse software (Michael Pusch, Genova, Italy). Recording pipettes pulled from borosilicate glass were filled with 3 M KCl and had resistances of 0.4-1 MΩ. Recordings were performed in ND96 solution containing (in mmol $L^{-1}$) 96 NaCl, 4 KCl, 1 CaCl₂, 1.8 MgCl2, 5 HEPES, pH 7.2–7.4 with 0.1% dimethylsulfoxide. For experiments with high extracellular potassium or rubidium concentrations, standard ND96 solution was replaced by 100 mM KCl and 100 mM RbCl solution, respectively. (*R*)-L3 was added to each bath solution to give a final concentration of 1 μM, from a 10 mM stock solution in DMSO. To determine the effects of (*R*)-L3 on wildtype and mutant $K_v7.1$ channels, currents were recorded by repetitive 7-second pulses to potentials of −100 mV to +60 mV, applied in 20 mV increments and then held for a 3-second tail pulse at −120 mV before returning to the holding potential of −80 mV. Mutations can have an influence on channel gating kinetics, plasma membrane expression and subsequently on the current amplitude. Wildtype and mutant channels expressed in *Xenopus laevis* were only used if the current at +40 mV was ≥0.5 μA, which is

robust enough for analyses. Throughout the manuscript, (R)-L3 effects on different KCNQ1 mutants are always discussed in comparison to the control recording at the same mutant in the absence of (R)-L3. Therefore, a correlation of plasma membrane expression and ionic current was not necessary.

$Co^{2+}$-assisted crosslinking based on the well-characterized affinity of transition metals to sulfhydryl groups of cysteines[46]. Formation of those metal-assisted bridges is only possible, if $Co^{2+}$ ion as well as sulfhydryl groups of two cysteines are in the right distance and angle to each other[47,48]. As a result of formation conformational changes can be restricted and subsequently protein function can be disturbed. For the $Co^{2+}$-assisted cysteine crosslinking oocytes were injected right before measurements with 2 mmol $L^{-1}$ CoCl2 and expressed channels were activated by +40 mV voltage pulse in the presence and absence of (R)-L3. The resulting current was normalized to oocytes expressing the same Kv7.1 variant without CoCl2 injection and without (R)-L3 application.

**Data Analysis**. Data were analyzed with the custom program Ana and GraphPad Prism 5.01 software (GraphPad Software, San Diego, California). All experimental results are presented as mean ± SEM with n specifying the number of independent experiments. Statistical analysis was performed with Student's $t$ test (unpaired) or ANOVA. The activation curves for wildtype and mutant Kv7.1 channels were determined from tail currents at hyperpolarized potentials by plotting the normalized peak tail current amplitudes versus the respective test potential. The resulting curves were fitted to a Boltzmann equation that gave an adequate fit with the two parameters $V_{1/2}$ (voltage of half-maximal activation) and the slope factor k. Activation kinetics was obtained by double-exponential fits to the currents yielding fast and slow time constants of activation. Deactivation rate constants were also obtained by double-exponential fits to the tail currents in the absence and presence of (R)-L3 and plotted against the applied membrane potential.

**Methods of voltage-clamp fluorometry (VCF)**. Oocytes were injected with 9.2 ng of cRNA encoding of psWT Kv7.1 (C214A/G219C/C331A) alone or simultaneously with 2.3 ng of cRNA encoding wt CiVSP. After incubation for 3–7 days at 18 C, the cells were labeled in high $K^+$ labeling solution (98 mmol $L^{-1}$ KCl, 1.8 mmol $L^{-1}$ CaCl2, 5 mmol $L^{-1}$ HEPES, pH 7.6) with 10 μM Alexa 488 $C_5$-maleimide (Molecular Probes) for 45 min on ice, washed with ND96 solution (96 mmol $L^{-1}$ NaCl, 4 mmol $L^{-1}$ KCl, 1.8 mmol $L^{-1}$ CaCl$_2$, 1 mmol $L^{-1}$ MgCl$_2$, 5 mmol $L^{-1}$ HEPES), and kept on ice until recording to minimize recycling of labeled channels. VCF recordings were carried out in ND96 solutions + / - (R)-L3. Fluorescence recordings were performed using a Leica DMLFS upright fluorescence microscope with a FITC filter cube. Emission from the animal pole was focused onto a pin 20 A photodiode (OSI Optoelectronics) and amplified using an EPC10 (HEKA) patch amplifier and analog filtered at 200 Hz. Simultaneous two-electrode voltage-clamp recordings were measured using a Dagan CA1-B amplifier in TEVC mode. Current and fluorescence signals were digitized at 1 kHz.

**Cell culture, transfection, photo-crosslinking, and cell lysis**. HEK293T cells were cultured in DMEM (Sigma-Aldrich) supplemented with 2 mmol $L^{-1}$ L-glutamine (Sigma-Aldrich), 100 U ml$^{-1}$ penicillin (Sigma-Aldrich), 10% fetal bovine serum (Biochrom) and 100 μg ml$^{-1}$ streptomycin (Sigma-Aldrich) at 37 °C / 5% $CO_2$. For transfection, cells were grown to 60–70% confluency in a 3.5 cm dish. For amber suppression experiments, the plasmid DNA ratio used for transfection was 1/1/0.2 (Kv7.1/suppressor tRNA/AzF aaRS). Transfection was performed using FuGene HD (Promega) according to the manufacturer's protocol. For the double transfection 625 ng Kv7.1 DNA, 625 ng suppressor tRNA DNA and 125 ng aaRS DNA were used per 3.5 cm dish. For transfection of wildtype (wt) Kv7.1, 1 μg of DNA was used per 3.5 cm dish. Non-transfected cells served as controls. Cells transfected with Kv7.1 amber mutants were grown in the presence of 0.5 mmol $L^{-1}$ AzF (Chem-Impex) in the medium. Cells expressing Kv7.1-M238AzF were harvested by trypsinization 24–48 h post-transfection, centrifuged, exposed to 254 nm UV-light for 3 min and pelleted again. The pellet was solubilized for 30 min at 4 °C and cell lysates were purified using the μMacs GFP isolation kit (Miltenyi Biotec) according to the manufacturer's protocol. SDS-PAGE was performed and proteins were transferred to nitrocellulose membranes. Following protein transfer, membranes were stained with Ponceau S solution (Sigma-Aldrich) and destained with Tris Buffered Saline (TBS: 20 mmol $L^{-1}$ Tris, 150 mmol $L^{-1}$ NaCl). For detection of Kv7.1, the membrane was immunoblotted with monoclonal Kv7.1 antibody (sc-365186, Santa Cruz Biotechnology) 1:200 in TBS-T (TBS + 0.1% Tween).

**Molecular modeling**. Molecular modeling was performed using YASARA 19 and OriginPro 2019 for data analysis. Kv7.1 structures of channel states AO, AC, and RC were adopted from Kuenze et al.[38]. Simulation parameters are given in SI Table 3. PDB files of different channel states are comprised of four identical KCNQ1 subunits named Mol A – D. Structure of (R)-L3 was constructed with YASARA 19 with defined stereochemistry and energy minimized using AMBER14 force field. Based on results from previous and currently performed mutagenesis studies, energy minimized (R)-L3 was manually docked between Mol A and Mol B in close proximity to all interacting amino acids from S4 (Mol B), S5 (Mol A) and

S6 (Mol A) followed by a second energy minimization using AMBER14[23]. For MD simulation in a lipid membrane, the provided YASARA 19 macro md_runmembrane.mcr was used and the following parameters were adapted: Membrane composition was set to 1/3 phosphatidyl-ethanolamine (PEA), 1/3 phosphatidyl-choline (PCH) and 1/3 phosphatidyl-serine (PSE). All polar head groups are 1-palmitoyl- and 2-oleoyl substituted. Simulation speed was set to fast (simulation time steps of 2*2.5 fs) and simulation time was set to 30 ns. Process of simulation was documented by screenshots every 0.25 ns leading to 120 screenshots in total for every simulation. RMSD, RMSF, and DCCM were calculated with provided YASARA macros md_analyze.mcr and md_analyzeres.mcr. Full simulation and analysis were conducted using AMBER14 force field.

**Statistics and reproducibility**. Where relevant, significance of mean differences for all data was analyzed using OriginPro 2021 by one-way-ANOVA and posthoc mean comparison Tukey test indicated by *$p$ values < 0.05, **$p$ values < 0.01, ***$p$ values < 0.001 and ns (not significant) for $p$ values > 0.05. Number of individual oocytes or simulations is given for every experiment in the main article, the figure legends, or the Supplementary information. Further, $n$ values can be reconstructed from Supplementary data for each figure panel.

**Reporting summary**. Further information on research design is available in the Nature Research Reporting Summary linked to this article.

## Data availability

Uncropped and unedited blot image for Fig. 5e is given as SI in the Supplementary Information. Source data for graphs and charts are given as supplementary data for each figure panel as excel sheets. Number of independent oocytes for kinetic parameters in Fig. 4 and $Co^{2+}$ measurements are given in SI Table 1 and SI Table 2. Further datasets generated during and/or analyzed during the current study are available from the corresponding author on reasonable request.

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

## Acknowledgements
The authors thank Geoffrey Abbott for providing $K_v$7.1 mutant channel constructs. This work was supported by R01HL126774, R01NS092570, and NS092570 from NIH to J.C. and J.o.S. and an AHA fellowship (11PRE5720009) to MZ.

## Author contributions
M.M., N.R., S.B., E.W., M.Z., L.Z., P.H., J.i.S., J.o.S., and Z.B. performed experiments, J.A.S., G.S. performed 3D simulations. Experiments were designed by N.S.S., J.C., B.W., J.A.S., and G.S. J.A.S. and G.S. wrote the manuscript with the help of M.M., E.W., N.S., N.S.S., B.W., N.D., M.D., and S.G.M.

## Funding

## Competing interests
The authors declare the following competing interests: J.i.S and J.C. are cofounders of a startup company VivoCor LLC, which is targeting IKs for the treatment of cardiac arrhythmia. Other authors declare they have no competing interests.
