## [Peer Review File · Communications Biology]

Reviewers' comments:

Reviewer #1 (Remarks to the Author):

The authors report a comprehensive analysis of the mechanism underlying KCNQ1 activation by the benzodiazepine, R-L3. R-L3 is a possible lead compound for drug development for KCNQ1 and IKs (KCNQ1-KCNE1) activators. KCNQ1 in complexes with KCNE1 or other KCNE subunits is important for the function of a wide range of tissues and cell types and its regulation by small molecules could lead to therapies for cardiac arrhythmia and various epithelial cell disorders.

The results of the study show that R-L3 probably interacts with the voltage sensor of KCNQ1 and also allosterically modulates the pore domain via interaction with S5 and S6. The team uses a multidisciplinary approach to investigate the binding site and mechanism of action. The work is novel, rigorous and will be of broad interest to ion channel physiologists and pharmacologists. I do not recommend any additional experiments as the study is comprehensive but I suggest additional discussion points:

1) R237A has large effects on KCNQ1 gating, increasing activation rate and/or slowing channel closure, leading to accumulation in an open state. Could the authors please discuss how the baseline effects of the various mutants govern their response to R-L3 and interpretation of the results.

2) Another compound (quercetin) was recently reported in *Comms Biology* to act via both the voltage sensor and pore of KCNQ1. It would be useful for the authors to compare their current work to these findings.

3) In the case of neuronal KCNQ channels, effects of some small molecules (retigabine) are reliant on by PIP2 while effects of others (Zn-pyrithione, GABA) are resistant to nominal PIP2 depletion. Is there a common theme in terms of ligand chemistry or binding site location that emerges from the available data with respect to PIP2 sensitivity?

Reviewer #2 (Remarks to the Author):

In the manuscript, Möller and colleagues describe the binding site of the benzodiazepine compound (R)-L3 on Kv7.1 homomeric channel structure and infer the putative molecular rearrangements that underlie the drug-induced alterations in the channel properties. (R)-L3 slows the gating kinetics of Kv7.1 channel and increases the amplitude of the current. Using different methods authors argue that the binding of the drug to the cleft between the voltage sensor domain (VSD), S4-S5-linker and pore domain affects the channels gating kinetics via perturbation of coupling between the VSD and pore domain. The interaction of (R)-L3 with the pore domain, in contrast, leads to allosteric modifications in the structure of the ionic pathway leading to a potentiation of the current amplitude. The results largely support the conclusion that (R)-L3 binding to VSD might be the cause of the modification of the channel gating kinetics. The mechanism of the drug action on current amplitude, in contrast, is less convincing – especially the electrophysiological part of the work. I encourage the authors to address the following points of concern:

Major points

1) Fig 2 A-C: the figure presentation and the corresponding description of the data in the results leave the reader wondering how the authors came to the conclusion in this set of experiments. Usage of the permeant ion Rb⁺ is a realistic strategy to discriminate the effect of (R)-L3 on the conductive properties of the channel pore (or the selectivity filter) from that of gating kinetics. However, the analysis and demonstration of the data (Fig 2 A-C) require substantial improvements.

Rb⁺ does not affect the tail current kinetics of Kv7.1 channel by itself (see Pusch et al., 2000),

which allows accurate estimation of the effect of permeant ions on current amplitudes in the presence and absence of (R)-L3. I suggest to compare the extent of the (R)-L3-induced potentiation of tail current observed in the presence of 100 mM K⁺ with that of at 100 mM Rb⁺. Remove data related to 96 Na⁺ (4 mM K⁺), since the external K⁺ has a significant effect on channel by itself (Larssen et al., 2011). Correction of the tail amplitudes for hook inactivation at both 100 mM K⁺ and 100 mM Rb⁺ is essential, since (R)-L3 significantly affects the hook-inactivation. The low expression of Kv7.1 will allow authors to avoid space clamp problems (Fig. 2B).

2) The estimation of the concentration-dependency for (R)-L3 effect in oocytes will be very helpful. The observed effect of the drug at 10 μ M (Fig 3B) is not comparable with that obtained at 1 μ M concentration presented throughout the manuscript.

3) The complex dynamics of steady-state amplitude during 1200s 10 μ M (R)-L3 application (Fig. 3B) raises two important questions: i) whether a similar effect is observed with 1 μ M concentration of compound? If yes, ii) to which time point of application the data presented in Fig 1-4 correspond? Some compounds affect the physicochemical properties of the cell membrane, which leads to unspecific currents. The application of the drug on non-expressing cells could serve as a control.

4) Fig 3B: Please, add a capture or color the symbols correspondingly.

5) Fig 3E: Due to poor quality of the figure 3E it is difficult to distinguish the GV and FV curves from each other. It is also unclear why certain data points (corresponding to 10 μ M (R)-L3 and CiVSP) acquire negative values. This is not readily inferred from the corresponding traces.

6) Fig. 4A: Please, including sample current traces for the important mutant M238A.

7) Fig 5. If I understood correctly, the authors state that M238 interacts with S5 pore of Kv7.1 channel in the open state of the channel. The proper control verifying such a state-dependent interaction will be the Western blot analysis at the hyperpolarized conditions (low external K⁺ conditions).

8) Fig. 6L: Please, provide the sample traces for W248 mutants.

9) Authors built the model of (R)-L3 – K7.1 channel complex "... by manual docking ...". Please, provide a brief and comprehensive methodological description how the model was assembled.

Minor points,

1) The references Nr. 23 and Nr. 48 are identical.

2) Fig. 5E: I271 in the text and I274 in the figure should be corrected to L271.

3) Fig 3F: Please, provide V_{1/2} and K values for Boltzmann fits and the corresponding statistics test results in legend or in the text.

4) Please, indicate the specific ANOVA test applied to the data in the legend of Fig. 4B

Reviewer #3 (Remarks to the Author):

The manuscript by Moller et al. describes the mechanism by which benzodiazepine (R)-L3 activates homomeric Kv7.1 channel. The authors use voltage-clamp recording and fluorometry combined with alanine scanning mutagenesis to identify several residues that mediate (R)-L3-induced current potentiation and effects on the activation and deactivation rates of Kv7.1 channel. Molecular dynamics (MD) simulations was also performed to identify hydrophobic residues that interact with the (R)-L3. Based on the movement correlation analyses in simulations, authors

propose that the (R)-L3 binding to the lower S4 and the S4-S5 linker results in the separation between the voltage sensor domain (VSD) and the pore domain by decreasing the movement correlation between the S4 and S5 and simultaneously increasing the correlation between S4 and S4-S5 linker and between the inner pore domain of S5 and S6.

The manuscript addresses an important question regarding the molecular actions of (R)-L3 in modulating Kv7.1 gating, considering the therapeutic potential of (R)-L3 in long QT syndrome-1, a devastating cardiac arrhythmia. However, the current manuscript has several major shortcomings that require attention. Significant revisions and additional experiments are needed to strengthen the author's conclusions and make this manuscript suitable for publication.

Major concerns:

1. In Figure 4, authors used alanine scanning mutagenesis to identify several residues that mediate (R)-L3-induced current potentiation and effects on the activation and deactivation rates of Kv7.1 channel. Many disease-causing mutations in S4 of voltage-gated potassium channels including Kv7 affect their protein and surface expression in oocytes and heterologous cells. What were the basal current level and surface expression of mutant channels? Are they comparable to the WT channels? These important controls are not presented in this data.

2. In line with the concern #1, the authors should describe their basal current expression and their relationship to their activation and deactivation kinetics in the absence of (R)-L3. This is relevant to the results in Figure 6 that identifies L236, M238, L239, V241, and W248 as (R)-L3 interacting residues. The I235A, R237A, and H240A mutations altered (R)-L3-induced current potentiation and kinetics, even though they are not binding to (R)-L3. How do authors explain this?

3. In Figure 1, (R)-L3 significantly increases fast and slow activation and deactivation time constants. In Figure 4B, do the kinetics in WT channels change significantly? No statistics (*) are shown. Does H240A and V241 significantly increase fast activation kinetics? Tau1 measurement of R237A is missing. There are only 1 sentence describing the effects of all tested mutations on (R)-L3-induced current potentiation and changes in kinetics. The description should be expanded.

3. Figure 5 (photo-crosslinking experiments) is not convincing. First of all, why does western blot with Kv7.1 antibody show multiple protein bands that are not Kv7.1 monomer, dimer, tri/tetramer? In the right 2 lanes of Figure 5E immunoblot, UV irradiation increases the protein band density of tri-tetramer of Kv7.1-M238Azido-Phe, and decreases the band density of Kv7.1-M238Azido-Phe monomer. However, I do not see any difference in Kv7.1-EGFP-photo-methionin expression upon UV irradiation. Due to this discrepancy, the authors should repeat these photo-crosslinking experiments and show statistical analysis of the western blot band intensities.

4. Additional comment on Figure 5. If M238-I271 inter subunit interaction was already reported from Ref 35 and 36, what is the purpose of this Figure, especially in relation to (R)-L3? Does this interaction necessary for (R)-L3? Can you test this?

5. In Figure 6, MD simulation has identified that (R)-L3 interacts with L236, M238, L239, and V241 in the lower S4 as well as W248 in the S4-S5 linker. How do W238F/A/R mutations alter currents and kinetics of Kv7.1 channels with and without (R)-L3?

6. The (R)-L3 interacting residues identified by the MD simulation (L236, M238, L239, and V241 in the lower S4 and W248 in the S4-S5 linker) should be tested for their roles in the VSD to the pore coupling by the voltage clamp fluorometry with or without (R)-L3. According to the authors' conclusions, one can speculate that if these mutations impair (R)-L3 binding, the effect of (R)-L3 on decoupling the VSD and the pore will be reduced. This experiment is crucial to support the modeling data and DCCM results presented in Figures 7-8.

7. In the Abstract, the authors state that "Summarizing, we provide structural and functional evidence for two independent Kv7.1 activating mechanisms by a single modulator mimicking PI(4,5)P2". I do not see any direct data nor sufficient discussion of the literature that supports the phrase "a single modulator mimicking PI(4,5)P2". The authors show that CiVSP-dependent PIP2

depletion does not affect (R)-L3-induced left shift of FV curve from the voltage-clamp fluorometry. How does this data translate to the above phrase?

8. Additional comment related to the above concern #7. Is there a competition between (R)-L3 and PIP2 in binding to Kv7.1? Is the MD simulation performed in the presence of PIP2 or absence of PIP2? In the methods, the membrane composition for MD simulation does not seem to include PIP2.

9. How is the action of (R)-L3 similar or different from the known action of PIP2? This should be discussed. Sun & MacKinnon 2019 paper should be cited to discuss the modulation of Kv7.1 channel gating by PIP2. The recent study by Liu et al. (PMID: 32678288) reports that a compound CP1, identified in silico based on the structures of both KCNQ1 and PIP2, can substitute for PIP2 to mediate VSD-pore coupling. This paper should be discussed.

Minor points.

1. The texts within Figures 1-3 and 6 are very small. Yellow texts in Figure 7 are hard to see.

2. Several typos are detected. One example is in the title (Phosphatidylinositol 4,5-bisphosphat). Another example is in line7 of the third paragraph in the Result section (voltage-dependend).

3. I understand that it is not easy to label all amino acid residues in Figures 7-8. However, if some of the residues are specifically mentioned in the main text (for example, in the last paragraph of the Result section, T312-K318, F339, V310-I313"), these specific residues should be labelled in X-axes of DCCMs. I suggest that authors make the DCCMs in Figures 7-8 bigger and include the missing residues.

4. The legend titles are missing from the Figures 6-8 legends.

Point-to-Point-Response to the Editor

Dear Professor Seebohm,

Your manuscript entitled "An Activator Locks Kv7.1 Channels Open by Electro-Mechanical Uncoupling and Allosterically Modulates its Pore with similarity to phosphatidylinositol-4,5-bisphosphat" has now been seen by 3 referees, whose comments are appended below. You will see from their comments copied below that while they find your work of potential interest, they have raised quite substantial concerns that must be addressed. In light of these comments, we cannot accept the manuscript for publication, but would be interested in considering a revised version that addresses these serious concerns.

In addition to other comments, we would specifically like you to include WB analysis at hyperpolarized condition as requested by R#2 and repeat photo-crosslinking experiments and test residues identified by MD with voltage clamp fluorometry with and without (R)-L3 as requested by R#3.

Reviewer #1 (Remarks to the Author):

The authors report a comprehensive analysis of the mechanism underlying KCNQ1 activation by the benzodiazepine, R-L3. R-L3 is a possible lead compound for drug development for KCNQ1 and IKs (KCNQ1-KCNE1) activators. KCNQ1 in complexes with KCNE1 or other KCNE subunits is important for the function of a wide range of tissues and cell types and its regulation by small molecules could lead to therapies for cardiac arrhythmia and various epithelial cell disorders. The results of the study show that R-L3 probably interacts with the voltage sensor of KCNQ1 and also allosterically modulates the pore domain via interaction with S5 and S6. The team uses a multidisciplinary approach to investigate the binding site and mechanism of action. The work is novel, rigorous and will be of broad interest to ion channel physiologists and pharmacologists. I do not recommend any additional experiments as the study is comprehensive but I suggest additional discussion points:

1) R237A has large effects on KCNQ1 gating, increasing activation rate and/or slowing channel closure, leading to accumulation in an open state. Could the authors please discuss how the baseline effects of the various mutants govern their response to R-L3 and interpretation of the results.

RESPONSE: We agree with reviewer #1, that mutations can have an influence on channel gating kinetics and subsequently on the current amplitude. Therefore, (R)-L3 effects on different KCNQ1 mutants are always discussed in comparison to the control measurements at the same mutant in absence of (R)-L3. In order to clarify this point, we include this information in the revised manuscript: “*Mutations can have an influence on channel gating kinetics, plasma membrane expression and subsequently on the current amplitude. Wild type and mutant channels expressed in *Xenopus laevis* were only used if the current at 40 mV was $\geq 0.5 \mu A$ which is robust enough for analyses. Throughout the manuscript, (R)-L3 effects on different KCNQ1 mutants are always discussed in comparison to the control recording at the same mutant in absence of (R)-L3. Therefore, a correlation of plasma membrane expression and ionic current was not necessary.*”

2) Another compound (quercetin) was recently reported in Comms Biology to act via both the voltage sensor and pore of KCNQ1. It would be useful for the authors to compare their current work to these findings.

RESPONSE: We thank the reviewer for this suggestion. A short comparison regarding the effects of quercetin and (R)-L3 is added to the manuscript.

3) In the case of neuronal KCNQ channels, effects of some small molecules (retigabine) are reliant on by PIP₂ while effects of others (Zn-pyrithione, GABA) are resistant to nominal PIP₂ depletion. Is there a common theme in terms of ligand chemistry or binding site location that emerges from the available data with respect to PIP₂ sensitivity?

RESPONSE: The sensitivity of a KCNQ modulator for PIP₂ presence/absence can be influenced by different parameters. Overlapping binding sites of the modulator and PIP₂ can lead to competition of the two molecules. If the affinity of the modulator is higher, PIP₂ can be displaced. However, if the modulator leads to similar conformational changes of the protein, the modulator can imitate PIP₂. On the other hand, if the modulator has only low affinity to his binding site that overlaps with a PIP₂ binding site, the modulator can become PIP₂ sensitive.

The sensitivity of a modulator for PIP₂ can also be caused by allosteric mechanisms. Binding of the modulator (or PIP₂) to a non-overlapping binding site can reduce the affinity for the binding of PIP₂ (or modulator) by the induction of conformational changes of the protein. It is also possible, that the modulator transduces its modulatory effect by residues, that are not affected by the conformational changes caused by PIP₂. Taken together, we cannot estimate common themes in terms of ligand chemistry or binding site location with sufficient certainty, since many aspects can influence the sensitivity for PIP₂.

Reviewer #2 (Remarks to the Author):

In the manuscript, Möller and colleagues describe the binding site of the benzodiazepine compound (R)-L3 on Kv7.1 homomeric channel structure and infer the putative molecular rearrangements that underlie the drug-induced alterations in the channel properties. (R)-L3 slows the gating kinetics of Kv7.1 channel and increases the amplitude of the current. Using different methods authors argue that the binding of the drug to the cleft between the voltage sensor domain (VSD), S4-S5-linker and pore domain affects the channels gating kinetics via perturbation of coupling between the VSD and pore domain. The interaction of (R)-L3 with the pore domain, in contrast, leads to allosteric modifications in the structure of the ionic pathway leading to a potentiation of the current amplitude. The results largely support the conclusion that (R)-L3 binding to VSD might be the cause of the modification of the channel gating kinetics. The mechanism of the drug action on current amplitude, in contrast, is less convincing – especially the electrophysiological part of the work. I encourage the authors to address the following points of concern:

Major points

1) Fig 2 A-C: the figure presentation and the corresponding description of the data in the results leave the reader wondering how the authors came to the conclusion in this set of experiments. Usage of the permeant ion Rb⁺ is a realistic strategy to discriminate the effect of (R)-L3 on the conductive properties of the channel pore (or the selectivity filter) from that of gating kinetics. However, the analysis and demonstration of the data (Fig 2 A-C) require substantial improvements.

RESPONSE: We revised the complete figure 2 and extended the analysis of the presented data. Further, we specify the results and discussion section made for this figure.

Rb⁺ does not affect the tail current kinetics of Kv7.1 channel by itself (see Pusch et al., 2000), which allows accurate estimation of the effect of permeant ions on current amplitudes in the presence and absence of (R)-L3. I suggest to compare the extent of the (R)-L3-induced potentiation of tail current observed in the presence of 100 mM K⁺ with that of at 100 mM Rb⁺. Remove data related to 96 Na⁺ (4 mM K⁺), since the external K⁺ has a significant effect on channel by itself (Larssen et al., 2011). Correction of the tail amplitudes for hook inactivation at both 100 mM K⁺ and 100 mM Rb⁺ is essential, since (R)-L3 significantly affects the hook-inactivation. The low expression of Kv7.1 will allow authors to avoid space clamp problems (Fig. 2B).

RESPONSE: We follow the advice of the reviewer. We compare the extent of the (R)-L3-induced potentiation of tail current observed in the presence of 100 mM K⁺ with that of at 100 mM Rb⁺. The data related to 96 Na⁺ were removed from the figure but mentioned in the text. Tail amplitudes corrected for hook inactivation as described by Seebohm et al., 2003 at both 100 mM K⁺ and 100 mM Rb⁺ was done and data is presented in Fig. 2.¹ As outlined by the reviewers general remarks above “the mechanism of the drug action on current amplitude is less convincing” we clearly state this concern in the discussion: “Indeed, the increment of corrected tail current amplitude by the modulator is dependent on the permeating ion (Figure 2C). However, direct comparison of ion effects is hampered since external K⁺ impacts onset of channel inactivation rendering interpretation less clear.”.

2) The estimation of the concentration-dependency for (R)-L3 effect in oocytes will be very helpful. The observed effect of the drug at 10 μM (Fig 3B) is not comparable with that obtained at 1 μM concentration presented throughout the manuscript.

RESPONSE: A dose response curve for (R)-L3 at Kv7.1 channels is now added as Figure 1D to the manuscript. For experiments presented in figure 3 a different Kv7.1 construct (Kv7.1_{VCF}) was used. Therefore, (R)-L3 activity can be altered and is not directly comparable. In our hands, the voltage clamp fluorometry with Kv7.1_{VCF} wt and especially mutant constructs showed relatively low signal to noise ratio and the adaptations were required to compensate for relatively low signals and clearly interpretable strong signals. Therefore, the concentration was elevated to 10 μM. As noted in the ms some mutant channels could not even be recorded under these conditions as the expressed too weakly.

3) The complex dynamics of steady-state amplitude during 1200s 10 μ M (R)-L3 application (Fig. 3B) raises two important questions: i) whether a similar effect is observed with 1 μ M concentration of compound? If yes, ii) to which time point of application the data presented in Fig 1-4 correspond? Some compounds affect the physicochemical properties of the cell membrane, which leads to unspecific currents. The application of the drug on non-expressing cells could serve as a control.

RESPONSE: The data presented in Figure 3 result from independent measurements with a different $K_v7.1$ construct ($K_v7.1_{VCF}$), that is exclusively optimized for the following voltage clamp fluorometry. Therefore, concentration of (R)-L3, expression rates as well as the applied pulse protocols differ from those used for the generated data in Fig 1-4. The voltage clamp fluorometry is a very sensitive technique with low signal to noise ratio, that needs highly accurate parameters, that are not always transferable to normal voltage clamp measurements.

4) Fig 3B: Please, add a capture or color the symbols correspondingly.

RESPONSE: We revised the complete Figure. Figure 3B represents the steady state currents at +60 mV under the presence of (R)-L3 from four individual measurements combined with the resulting mean \pm SEM. We introduced a color code and a more detailed description in the figure legend to specify the five traces.

5) Fig 3E: Due to poor quality of the figure 3E it is difficult to distinguish the GV and FV curves from each other. It is also unclear why certain data points (corresponding to 10 μ M (R)-L3 and CiVSP) acquire negative values. This is not readily inferred from the corresponding traces.

RESPONSE: We revised the complete Figure 3 to increase the visibility. The negative data, that were previously presented, resulted from a scaling error. We apologize for this mistake and correct the figure accordingly.

6) Fig. 4A: Please, including sample current traces for the important mutant M238A.

RESPONSE: We added the sample trace for M238A to the supporting information figure 3.

7) Fig 5. If I understood correctly, the authors state that M238 interacts with S5 pore of Kv7.1 channel in the open state of the channel. The proper control verifying such a state-dependent interaction will be the Western blot analysis at the hyperpolarized conditions (low external K⁺ conditions).

RESPONSE: Western Blot analyses under hyperpolarized conditions are a proper method for verifying the interaction between M238 and I274, however, they are also error-prone and less controllable. The coworker originally producing these data left the lab and within the last month we did not succeed to set up the system again. To address the point raised by the reviewer we choose a new approach to verify the interaction using Co²⁺-sensitive double mutant M238C/I274C (revised Fig 5). Cysteines can bind Co²⁺ ions and form Co²⁺-mediated cysteine bridges if the residues are in the right orientation and distance to each other. While the wildtype (revised Figure 5) as well as the single mutants M238C and I274C (Supporting information) are insensitive to Co²⁺, the double mutant is significantly inhibited by Co²⁺ underlining the interdomain interaction between these two amino acids.

8) Fig. 6L: Please, provide the sample traces for W248 mutants.

RESPONSE: We added the sample trace for M238A to the supporting information figure 3.

9) Authors built the model of (R)-L3 – K7.1 channel complex "... by manual docking ...". Please, provide a brief and comprehensive methodological description how the model was assembled.

RESPONSE: A brief and comprehensive methodological description is given in the materials and methods part. No additional procedures were conducted. For better understanding the term "manual docking" was replaced by the following term: After construction, (R)-L3 was placed in close proximity to all interacting amino acids previously identified by mutagenesis² from S4 (Mol B), S5 (Mol A) and S6 (Mol A) followed by a second energy minimization using force field AMBER14.

Minor points,

1) The references Nr. 23 and Nr. 48 are identical.

RESPONSE: We correct the references.

2) Fig. 5E: I271 in the text and I274 in the figure should be corrected to L271.

RESPONSE: We apologize for the confusion. (R)-L3 interacts directly with L271. On the other hand, M238 interacts with (R)-L3 and I274. The analysis made in Fig 5E refer to the inter-domain interaction between M238 and I274. We clarified the mentioned interaction in the text.

3) Fig 3F: Please, provide $V_{1/2}$ and K values for Boltzmann fits and the corresponding statistics test results in legend or in the text.

RESPONSE: We added the values to the text of the manuscript.

4) Please, indicate the specific ANOVA test applied to the data in the legend of Fig. 4B

RESPONSE: data were analyzed by one-way-ANOVA followed by posthoc mean comparison Tukey test. Statistical tests are now clearly indicated.

Reviewer #3 (Remarks to the Author):

The manuscript by Moller et al. describes the mechanism by which benzodiazepine (R)-L3 activates homomeric Kv7.1 channel. The authors use voltage-clamp recording and fluorometry combined with alanine scanning mutagenesis to identify several residues that mediate (R)-L3-induced current potentiation and effects on the activation and deactivation rates of Kv7.1 channel. Molecular dynamics (MD) simulations were also performed to identify hydrophobic residues that interact with the (R)-L3. Based on the movement correlation analyses in simulations, authors propose that the (R)-L3 binding to the lower S4 and the S4-S5 linker results in the separation between the voltage sensor domain (VSD) and the pore domain by decreasing the movement correlation between the S4 and S5 and simultaneously increasing the correlation between S4 and S4-S5 linker and between the inner pore domain of S5 and S6. The manuscript addresses an important question regarding the molecular actions of (R)-L3 in modulating Kv7.1 gating, considering the therapeutic potential of (R)-L3 in long QT syndrome-1, a devastating cardiac arrhythmia. However, the current manuscript has several major shortcomings that require attention. Significant revisions and additional experiments are needed to strengthen the author's conclusions and make this manuscript suitable for publication.

Major concerns:

1. In Figure 4, authors used alanine scanning mutagenesis to identify several residues that mediate (R)-L3-induced current potentiation and effects on the activation and deactivation rates of Kv7.1 channel. Many disease-causing mutations in S4 of voltage-gated potassium channels including Kv7 affect their protein and surface expression in oocytes and heterologous cells. What were the basal current level and surface expression of mutant channels? Are they comparable to the WT channels? These important controls are not presented in this data.

RESPONSE: We agree with the reviewer, that mutations in the S4 helix affect the surface expression, the current level and the kinetic of the channel. Indeed, several studies indicate reduced function and / or decreased plasma membrane expression.^{3,4} Therefore, we do not directly compare the effect of (R)-L3 on a mutant channel to the effect on wildtype channels. To examine the influence of a residue on the (R)-L3 activation mechanism, we analyze compound effects on the current amplitude and kinetic parameters upon (R)-L3 application to the same mutant in the same oocyte under identical conditions. Therefore, the (R)-L3 effects are determined in identical setting (mutant channel in the same oocyte) in an internal control. Sufficient expression of the respective mutant current is taken into account. For clarity, we include the following text in the manuscript: “*Mutations can have an influence on channel gating kinetics, plasma membrane expression and subsequently on the current amplitude. Wild type and mutant channels expressed in *Xenopus laevis* were only used if the current at 40 mV was $\geq 0.5 \mu A$ which is robust enough for analyses. Throughout the manuscript, (R)-L3 effects on different *KCNQ1* mutants are always discussed in comparison to the control recording at the same mutant in absence of (R)-L3. Therefore, a correlation of plasma membrane expression and ionic current was not necessary.*”

2. In line with the concern #1, the authors should describe their basal current expression and their relationship to their activation and deactivation kinetics in the absence of (R)-L3. This is relevant to the results in Figure 6 that identifies L236, M238, L239, V241, and W248 as (R)-L3 interacting residues. The I235A, R237A, and H240A mutations altered (R)-L3-induced current potentiation and kinetics, even though they are not binding to (R)-L3. How do authors explain this?

RESPONSE: We agree that sufficient basal current expression is relevant for the identification of interacting residues. To differentiate the mutant current from the leak currents / background sufficient expression has to be present. In order to address this point we include the following text in the manuscript: “*Mutations can have an influence on channel gating kinetics, plasma membrane expression and subsequently on the current amplitude. Wild type and mutant channels expressed in *Xenopus laevis* were only used if the current at 40 mV was $\geq 0.5 \mu A$, which is robust enough for analyses. Throughout the manuscript, (R)-L3 effects on different *KCNQ1* mutants are always discussed in comparison to the control recording at the same mutant in absence of (R)-L3. Therefore, a correlation of plasma membrane expression and ionic current was not necessary.*” Based on this statement it is now clear that current amplitude as well as kinetic parameters in presence of (R)-L3 can be compared to the same parameters resulting from measurements from the same mutant in absence of (R)-L3. We provided these comparisons in Figure 4. The presented data were collected by constantly applying the pulse protocol to a mutant expressing oocyte in absence of (R)-L3 until a steady-state was reached. After this, wash-in of (R)-L3 was performed and constant application of the pulse protocol until the new steady-state was reached. The data were analyzed and compared separately for every oocyte. With this procedure we can clearly distinguish between effects resulting from the mutation and effects resulting from application of (R)-L3.

We thank the reviewer for the comment in regard to figure 6: No direct effect between the side chains of I235, R237 and H240 exist in this model. This can be explained by two different aspects. First, modelling of different states can clearly alters the orientation of residue side chains. Moreover, *in silico* modelling often assumes strong rigidity of a protein that is not in strict accordance with the natural dynamic behavior of the protein. Therefore, *in silico* data can only provide an idea of the natural dynamic behavior. Secondly, in a densely packed protein region like the pore domain-voltage sensor interface allosteric coupling is highly relevant. Even if direct interaction between a modulator and residues inside the protein were not observed, the allosteric coupling, that is responsible to transfer the conformational changes generated by modulator binding through the protein region, will generate altered modulator behavior.

3. In Figure 1, (R)-L3 significantly increases fast and slow activation and deactivation time constants. In Figure 4B, do the kinetics in WT channels change significantly? No statistics (*) are shown. Does H240A and V241 significantly increase fast activation kinetics? Tau1 measurement of R237A is missing. There are only 1 sentence describing the effects of all tested

mutations on (R)-L3-induced current potentiation and changes in kinetics. The description should be expanded.

RESPONSE: We revised the whole figure and added the missing statistics. The slow activation as well as deactivation time constants are significantly increased in the wildtype by the application of (R)-L3. The mutant R237A dramatically altered the channel gating properties. Therefore, no discrimination between fast and slow component was possible and a single exponential fit was used exclusively for this mutant. We added a short statement regarding this problem of mathematical description of mutant kinetic behavior to the text. Further, we extended the description of the whole figure in the text.

3. Figure 5 (photo-crosslinking experiments) is not convincing. First of all, why does western blot with Kv7.1 antibody show multiple protein bands that are not Kv7.1 monomer, dimer, tri/tetramer? In the right 2 lanes of Figure 5E immunoblot, UV irradiation increases the protein band density of tri-tetramer of Kv7.1-M238Azido-Phe, and decreases the band density of Kv7.1-M238Azido-Phe monomer. However, I do not see any difference in Kv7.1-EGFP-photo-methionin expression upon UV irradiation. Due to this discrepancy, the authors should repeat these photo-crosslinking experiments and show statistical analysis of the western blot band intensities.

RESPONSE: Similar to other antibodies, the used antibody shows unspecific binding, that could not be eradicate by the use of different blotting protocols. In our hand this relatively specific antibody yields the best just-acceptable results for KCNQ1 staining out of 6 tested antibodies we have tested in the last 20 years on this channel. However, as the reviewer raises concerns regarding this approach, we develop a second approach to prove confirming evidence of the interdomain interaction between M238 and I274 that can clearly be detected in the two published structural models published by the MacKinnon group.^{5,6} Cysteines can build Co^{2+} -assisted bridges, if the side chains are in the right distance and orientation to each other. TEVC measurements in presence of Co^{2+} ions with the double mutant M238C/I274C could clearly verify the assumed interaction between these two residues (revised Fig. 5) since the wildtype as well as the single mutants are not sensitive to Co^{2+} (revised Fig. 5 and Supporting information figure 2). We are confident that these novel data together with the photo-crosslinking experiments and the structural data provided by the MacKinnon group provide hard evidence for the proposed interdomain interaction between M238 and I274.

4. Additional comment on Figure 5. If M238-I271 inter subunit interaction was already reported from Ref 35 and 36, what is the purpose of this Figure, especially in relation to (R)-L3? Does this interaction necessary for (R)-L3? Can you test this?

RESPONSE: M238 is interacting with (R)-L3 and the residue I274 from the pore domain. The coupling of these two amino acids is important for the interdomain coupling of the VSD and the PD. We showed, that (R)-L3 directly interacts with M238 leading to reorientation of the residue side chain, that reduces the interaction with I274. Using the Co^{2+} -assisted cysteine approach we could clearly verify the importance for channel activation for the interaction between M238 and I274. However, application of (R)-L3 to the double mutant M2C38/I274C in presence of Co^{2+} is able to relieve from the inhibition caused by Co^{2+} -bridging confirming that decoupling effects of the PD from VSD by (R)-L3 is involved in (R)-L3 mediated activation. Therefore, these results indicate that the interaction is very relevant for (R)-L3 activation.

5. In Figure 6, MD simulation has identified that (R)-L3 interacts with L236, M238, L239, and V241 in the lower S4 as well as W248 in the S4-S5 linker. How do W238F/A/R mutations alter currents and kinetics of Kv7.1 channels with and without (R)-L3?

RESPONSE: We thank the reviewer for this suggestion and added sample traces as well as the τ -values for activation and deactivation for all W248 mutants to the supporting information figure 3.

6. The (R)-L3 interacting residues identified by the MD simulation (L236, M238, L239, and V241 in the lower S4 and W248 in the S4-S5 linker) should be tested for their roles in the VSD to the pore coupling by the voltage clamp fluorometry with or without (R)-L3. According to the authors' conclusions, one can speculate that if these mutations impair (R)-L3 binding, the effect of (R)-L3 on decoupling the VSD and the pore will be reduced. This experiment is crucial to support the modeling data and DCCM results presented in Figures 7-8.

RESPONSE: We performed additional voltage clamp fluorometry measurements for all mentioned mutants. However, voltage clamp fluorometry needs high expression rates, that can not be obtained for all mutants, since mutations can have an influence on the surface expression

rates. Therefore, due to the weak expression and resulting low signal to noise ratio for the other mutants, we could not achieve suitable experimental data for the other mutants. Data for the mutants with sufficient expression for analysis were added to the supporting information figure 4.

7. In the Abstract, the authors state that “Summarizing, we provide structural and functional evidence for two independent Kv7.1 activating mechanisms by a single modulator mimicking PI(4,5)P₂”. I do not see any direct data nor sufficient discussion of the literature that supports the phrase “a single modulator mimicking PI(4,5)P₂”. The authors show that CiVSP-dependent PIP₂ depletion does not affect (R)-L3-induced left shift of FV curve from the voltage-clamp fluorometry. How does this data translate to the above phrase?

RESPONSE: We agree with the reviewer that this point is weakly covered in the manuscript. We follow his criticism and excluded the sentences in the abstract as well as the results and discussion section for the CiVSP measurements as PI(4,5)P₂ similarities are not a central subject of this manuscript.

8. Additional comment related to the above concern #7. Is there a competition between (R)-L3 and PIP₂ in binding to Kv7.1? Is the MD simulation performed in the presence of PIP₂ or absence of PIP₂? In the methods, the membrane composition for MD simulation does not seem to include PIP₂.

RESPONSE: All simulations are performed without bound PIP₂. A competition between PIP₂ and (R)-L3 can be nearly excluded, since the effects of (R)-L3 are independent from the presence of PIP₂ and the binding sites for PIP₂ and (R)-L3 do not largely overlap. To ensure this, we performed an overlay of our used structures with the previous published structure of the KCNQ1/KCNE3-CaM complex with PIP₂ (See picture below).

Therefore, we remove the PIP₂ related data and speculations from the manuscript as mentioned above.

9. How is the action of (R)-L3 similar or different from the known action of PIP₂? This should be discussed. Sun & MacKinnon 2019 paper should be cited to discuss the modulation of Kv7.1 channel gating by PIP₂. The recent study by Liu et al. (PMID: 32678288) reports that a compound CP1, identified *in silico* based on the structures of both KCNQ1 and PIP₂, can substitute for PIP₂ to mediate VSD-pore coupling. This paper should be discussed.

RESPONSE: We thank the reviewer for this good suggestion. Following this advice, we added a brief discussion of the two papers to the manuscript.

Minor points.

1. The texts within Figures 1-3 and 6 are very small. Yellow texts in Figure 7 are hard to see.

RESPONSE: We enlarged the labels in figures 1-3 as well as the complete Figures 6 and 7 to improve the visibility.

2. Several typos are detected. One example is in the title (Phosphatidylinositol 4,5-bisphosphat). Another example is in line7 of the third paragraph in the Result section (voltage-dependend).

RESPONSE: We corrected the typos.

3. I understand that it is not easy to label all amino acid residues in Figures 7-8. However, if some of the residues are specifically mentioned in the main text (for example, in the last paragraph of the Result section, T312-K318, F339, V310-I313”), these specific residues should be labelled in X-axes of DCCMs. I suggest that authors make the DCCMs in Figures 7-8 bigger and include the missing residues.

RESPONSE: We extended the figure size and increase the visibility of the single amino acids. However, it is not possible to label all single amino acids with the name. Thus, small indicators between the named amino acids can be used to identify the exact position of the exact amino acids in the text.

4. The legend titles are missing from the Figures 6-8 legends.

RESPONSE: The legend titles for the Figures 6-8 are added.

References

1. Seebohm, G. Tight coupling of rubidium conductance and inactivation in human KCNQ1 potassium channels. *J. Physiol.* **552**, 369–378 (2003).
2. Seebohm, G., Pusch, M., Chen, J. & Sanguinetti, M. C. Pharmacological Activation of Normal and Arrhythmia-Associated Mutant KCNQ1 Potassium Channels. *Circ. Res.* **93**, 941–947 (2003).
3. Henrion, U., Strutz-Seebohm, N., Duszenko, M., Lang, F. & Seebohm, G. Long QT Syndrome-associated Mutations in the Voltage Sensor of IKs Channels. *Cell. Physiol. Biochem.* **24**, 11–16 (2009).
4. Huang, H. *et al.* Mechanisms of KCNQ1 channel dysfunction in long QT syndrome involving voltage sensor domain mutations. *Sci. Adv.* **4**, eaar2631 (2018).
5. Sun, J. & MacKinnon, R. Cryo-EM Structure of a KCNQ1/CaM Complex Reveals

Insights into Congenital Long QT Syndrome. *Cell* **169**, 1042-1050.e9 (2017).

6. Sun, J. & MacKinnon, R. Structural Basis of Human KCNQ1 Modulation and Gating. *Cell* **180**, 340-347.e9 (2020).

Reviewers' comments:

Reviewer #2 (Remarks to the Author):

The authors have addressed most of my concerns and have added substantially to the manuscript. The manuscript has been significantly improved and the presentation of the data in figures became more discernible. Nevertheless, the following issues still remain unclear.

1) As mentioned before, the kinetic properties of the inward tail currents as well as the steady-state gating parameters of homomeric KCNQ1 channels at high external Rb⁺ are not significantly different from those at equimolar external K⁺ (Pusch et al., 2000). The results presented in revised Fig. 2 are largely consistent with this concept despite ~10 mV depolarizing shift of V_{1/2} in Rb⁺. Thus, Fig. 2 clearly shows that the potentiating effect of (R)-L3 (at concentration 1 μM) on inward current amplitude does not depend on the permeating ion. Authors argue that the comparison with previous data (at low external K⁺) is difficult due to the effect of external permeating ions on outward current amplitude and tail kinetics. So, the question arises: what does this set of experiments show?

On the other hand, an early study from authors revealed several mutations in the pore region that dramatically reduce (R)-L3 sensitivity of the channel. Authors could discuss the results shown in Fig. 2 together with previous mutagenesis results related to the pore residues. I suggest the following context for such a discussion. Almost threefold increase of inward current amplitude by Rb⁺ relative to K⁺ was earlier explained by unblocking of the flicker state of the channel—attributable either to selectivity filter or to the cytoplasmic activation gate. The functional effect of (R)-L3 on these structures can be discussed taking into consideration also the MD simulation data.

It would be advantageous to have an identical scale for Fig. 2F and 2G for easy comparison of deactivation time constants for different conditions.

2) The results of new experiments related to the intracellular injection of CoCl₂ are poorly described and presented. Original traces for each mutant with and without CoCl₂ injection should be included in the manuscript. This set of experiments does not provide evidence for the state-dependency of interaction, since intracellular Co²⁺ is present at both depolarized and hyperpolarized states of the voltage sensor. Co²⁺-bridge could be formed at hyperpolarized conditions as well but may affect the current amplitude at depolarized potentials. Different scenarios for such an effect of Co²⁺-bridge on current amplitude are imaginable.

3) The sentence "The data are also consistent with previous findings characterizing M238 as key residue for VSD coupling with the pore 34,35" should be removed, since these publications (indexed as "34" and "35") do not include description or mutational data that characterizes M238 residue as important player of coupling.

Reviewer #3 (Remarks to the Author):

The manuscript by Moller et al. describes the mechanism by which benzodiazepine derivative (R)-L3 activates homomeric Kv7.1 channel.

The revised manuscript has addressed all specific comments made in my earlier review by conducting additional experiments (such as VCF) and statistical analyses, and revising both texts and figures. These revisions significantly strengthened the authors' conclusion that the (R)-L3 potentiates the current through Kv7.1 channel by allosterically modulating its pore domain and by stabilizing its voltage sensor at the activate state.

2ND RESPONSE TO REVIEWER #2 AND #3

The authors have addressed most of my concerns and have added substantially to the manuscript. The manuscript has been significantly improved and the presentation of the data in figures became more discernible. Nevertheless, the following issues still remain unclear.

1) As mentioned before, the kinetic properties of the inward tail currents as well as the steady-state gating parameters of homomeric KCNQ1 channels at high external Rb⁺ are not significantly different from those at equimolar external K⁺ (Pusch et al., 2000). The results presented in revised Fig. 2 are largely consistent with this concept despite ~10 mV depolarizing shift of V_{1/2} in Rb⁺. Thus, Fig. 2 clearly shows that the potentiating effect of (R)-L3 (at concentration 1 μM) on inward current amplitude does not depend on the permeating ion. Authors argue that the comparison with previous data (at low external K⁺) is difficult due to the effect of external permeating ions on outward current amplitude and tail kinetics. So, the question arises: what does this set of experiments show?

On the other hand, an early study from authors revealed several mutations in the pore region that dramatically reduce (R)-L3 sensitivity of the channel. Authors could discuss the results shown in Fig. 2 together with previous mutagenesis results related to the pore residues. I suggest the following context for such a discussion. Almost threefold increase of inward current amplitude by Rb⁺ relative to K⁺ was earlier explained by unblocking of the flicker state of the channel attributable either to selectivity filter or to the cytoplasmic activation gate. The functional effect of (R)-L3 on these structures can be discussed taking into consideration also the MD simulation data.

RESPONSE: We thank reviewer #2 for the suggestions and revised the section regarding the results from Fig. 2 again. In response, we added a short comparison with the early study as suggested by the reviewer. Although the permeating ion (K⁺ / Rb⁺) does only slightly but not significantly influence the effects of (R)-L3 like the shift for V_{1/2}, the compounds effects are clearly modulated by different extracellular ion concentrations (96 mM Na⁺ / 100 mM K⁺ / 100 mM Rb⁺). The previous identified amino acids in the lower S5 and S6 helix, that are crucial for high (R)-L3 activity, clearly influence the pore domain in a direct manner. We followed the reviewers suggestion and added a new depiction of KCNQ1 showing these previous identified amino acids as Figure 2H to the manuscript.

It would be advantageous to have an identical scale for Fig. 2F and 2G for easy comparison of deactivation time constants for different conditions.

RESPONSE: We rescaled the figure panels 2F and 2G to the identical scale.

2) The results of new experiments related to the intracellular injection of CoCl_2 are poorly described and presented.

RESPONSE: We added the following detailed description for Co^{2+} /cysteine experiments to the materials and methods, that provide literature references as well as a mechanistic explanation: *Co^{2+} -assisted crosslinking based on the well characterized affinity of transition metals to sulfhydryl groups of cysteines.¹ Formation of those metal assisted bridges is only possible, if Co^{2+} ion as well as sulfhydryl groups of two cysteines are in the right distance and angle to each other.^{2,3} As a result of formation conformational changes can be restricted and subsequently protein function can be disturbed. For the Co^{2+} -assisted cysteine crosslinking oocytes were injected right before measurements with 2 mM CoCl_2 and expressed channels were activated by +40 mV voltage pulse in presence and absence of (R)-L3. The resulting current was normalized to oocytes expressing the same Kv7.1 variant without CoCl_2 injection and without (R)-L3 application.*

Original traces for each mutant with and without CoCl_2 injection should be included in the manuscript.

RESPONSE: We followed the reviewers advice and include sample traces in the supporting information.

This set of experiments does not provide evidence for the state-dependency of interaction, since intracellular Co^{2+} is present at both depolarized and hyperpolarized states of the voltage sensor. Co^{2+} -bridge could be formed at hyperpolarized conditions as well but may affect the current amplitude at depolarized potentials. Different scenarios for such an effect of Co^{2+} -bridge on current amplitude are imaginable.

RESPONSE: We revised the statement regarding the state-dependency to clarify, that this is our hypothesis, which is in good accordance with our experimental data as well as the actual

literature and published structures by Sun and MacKinnon.⁴ The published Cryo-EM structure (PDB 6UZZ) shows the KCNQ1 channel in a “decoupled” configuration with activated voltage sensors and a closed pore.⁴ The voltage sensor adopts the activated conformation by an upward movement and a rotation of the S1-S4 helix bundle, that reduces the distance between S4 and S5.⁵ In this structure M238 and I274 interact via hydrophobic interactions. If the VSD adopts the non-activated state, the previous described movement will be reversed resulting in a pronounced separation of S4 and S5. Consequently, interactions between M238 and I274 are not possible (See Figure 1).

Figure 1: Depiction of cryo-EM structure PDB ID 6UZZ showing KCNQ1 with an activated VSD and a closed Pore.

At early stages of the project, we tried to access the state-dependent interaction by performing crosslinking western blot using K_v7.1 M238AMBER in depolarizing high K⁺ as well as low

K⁺ buffer as a control. If the interaction is favored in the activated state under depolarizing conditions, a strong increment of multimer together with a pronounced decrement of monomer fraction should be observed in comparison to non-irradiated control. However, some unspecific binding as well as low band intensity makes it challenging to generate robust results. In general, crosslink reactions using unnatural amino acids are very sensitive and even under perfect conditions the overall crosslink percentage can be very low. After intensive protocol optimization for crosslinking and blotting, we achieved a clean western blot. This blot showed a more pronounced increase of multimer fraction in high K⁺ compared to the increase in low K⁺ buffer (Figure 2).

Figure 2: Western blotting using K_v7.1AMBER + AzF to examine state dependency.

3) The sentence "The data are also consistent with previous findings characterizing M238 as key residue for VSD coupling with the pore 34,35" should be removed, since these

publications (indexed as "34" and "35") do not include description or mutational data that characterizes M238 residue as important player of coupling.

RESPONSE: We rephrased the sentence. With our data we showed, that M238 is a crucial part of the coupling mechanism between VSD and PD, that is described in these two references.

Reviewer #3 (Remarks to the Author):

The manuscript by Moller et al. describes the mechanism by which benzodiazepine derivative (R)-L3 activates homomeric Kv7.1 channel.

The revised manuscript has addressed all specific comments made in my earlier review by conducting additional experiments (such as VCF) and statistical analyses, and revising both texts and figures. These revisions significantly strengthened the authors' conclusion that the (R)-L3 potentiates the current through Kv7.1 channel by allosterically modulating its pore domain and by stabilizing its voltage sensor at the activate state.

RESPONSE: We thank the reviewer for the helpful suggestions to improve the manuscript.

REFERENCES

1. Dilger, J. M., Glover, M. S. & Clemmer, D. E. A Database of Transition-Metal-Coordinated Peptide Cross-Sections: Selective Interaction with Specific Amino Acid Residues. *J. Am. Soc. Mass Spectrom.* **28**, 1293–1303 (2017).
2. MacColl, R. *et al.* Interrelationships among biological activity, disulfide bonds, secondary structure, and metal ion binding for a chemically synthesized 34-amino-acid peptide derived from α -fetoprotein. *Biochim. Biophys. Acta - Gen. Subj.* **1528**, 127–134 (2001).
3. Sóvágó, I., Kiss, T., Várnagy, K. & Révész, B. D.-L. Cobalt(II) and zinc(II) complexes of cysteine containing dipeptides. *Polyhedron* **7**, 1089–1093 (1988).
4. Sun, J. & MacKinnon, R. Structural Basis of Human KCNQ1 Modulation and Gating. *Cell* **180**, 340-347.e9 (2020).

5. Catacuzzeno, L., Sforza, L. & Franciolini, F. Voltage-dependent gating in K channels: experimental results and quantitative models. *Pflügers Arch. - Eur. J. Physiol.* **472**, 27–47 (2020).

REVIEWERS' COMMENTS:

Reviewer #2 (Remarks to the Author):

In the revised manuscript, the authors addressed all my previous points of concern. The manuscript has been significantly improved. I hope that the reader will have an opportunity to see the original current traces for Co₂+ bridging experiments in the revised supplement which is not available for me. The authors left in the manuscript the references 34 and 35, which are absolutely out of context and have nothing to do with discussion of the section. Except for these minor points other points raised in the previous review are properly addressed.

3rd response to the Reviewers

Reviewer #2 (Remarks to the Author):

In the revised manuscript, the authors addressed all my previous points of concern. The manuscript has been significantly improved. I hope that the reader will have an opportunity to see the original current traces for Co₂+ bridging experiments in the revised supplement which is not available for me.

RESPONSE: We added sample current traces for all conditions to the Supplementary Information. They can be found in SI Figure 2.

The authors left in the manuscript the references 34 and 35, which are absolutely out of context and have nothing to do with discussion of the section. Except for these minor points other points raised in the previous review are properly addressed.

RESPONSE: We have adopted the manuscript accordingly and thank the reviewer for the helpful suggestions to improve our manuscript.